# Phytochemical Profile, Antioxidant Capacity, and Photoprotective Potential of Brazilian *Humulus Lupulus*

**DOI:** 10.3390/ph18081229

**Published:** 2025-08-20

**Authors:** Gabriela Catuzo Canonico Silva, Fabiana Pereira Alves da Silva, Gabriel Augusto Rodrigues Beirão, José Júnior Severino, Mariane de Almeida Machado, Marina Pereira da Silva Bocchio Barbosa, Giulia Boito Reyes, Max Emerson Rickli, Ana Daniela Lopes, Ezilda Jacomassi, Maria Graciela Iecher Faria Nunes, João Paulo Francisco, Beatriz Cervejeira Bolanho Barros, Juliana Silveira do Valle, José Eduardo Gonçalves, Zilda Cristiani Gazim

**Affiliations:** 1Graduate Program in Biotechnology Applied to Agriculture, Universidade Paranaense, Umuarama 87502-210, Brazil; gabriela.canonico@edu.unipar.br (G.C.C.S.); fabiana.alves@edu.unipar.br (F.P.A.d.S.); gabriel.beirao@edu.unipar.br (G.A.R.B.); jose.258613@edu.unipar.br (J.J.S.); gracielaiecher@prof.unipar.br (M.G.I.F.N.); jsvalle@prof.unipar.br (J.S.d.V.); 2Graduate Program in Animal Science with Emphasis on Bioactive Products, Universidade Paranaense, Umuarama 87502-210, Brazil; mariane.machado@edu.unipar.br; 3Graduate Program in Medicinal and Phytotherapeutic Plants in Primary Care, Universidade Paranaense, Umuarama 87502-210, Brazil; marina.silva.81@edu.unipar.br (M.P.d.S.B.B.); ezilda@prof.unipar.br (E.J.); 4Graduate Program in Clean Technologies, UniCesumar, Maringá 87050-390, Brazil; giuliaboito18@gmail.com (G.B.R.); jose.goncalves@unicesumar.edu.br (J.E.G.); 5Department of Zootecnia, Universidade Estadual de Maringá, Maringá 87020-900, Brazil; merickli@uem.br; 6FMC Agrícola, Paulinia 13140-000, Brazil; anadanilopes1@gmail.com; 7Department of Agricultural Sciences, Universidade Estadual de Maringá, Campus Umuarama, Umuarama 87020-900, Brazil; jpfrancisco2@uem.br; 8Postgraduate Program in Sustainability, Universidade Estadual de Maringá, Campus Umuarama, Umuarama 87020-900, Brazil; bcbolanhobarros@uem.br; 9Cesumar Institute of Science, Technology and Innovation, UniCesumar, Maringá 87050-390, Brazil

**Keywords:** Brazilian hops, sun protection, rutin, protocatechuic acid, ABTS, female inflorescences, flavonoids, total phenolics, FRAP, DPPH

## Abstract

**Background and Objectives**: The cultivation of *Humulus lupulus* has been encouraged in Brazil, despite the country’s climate generally being unfavorable for its development. This study aimed to evaluate the chemical composition, antioxidant activity, and photoprotective potential of four *H. lupulus* varieties, Cascade, Columbus, Comet, and Nugget, cultivated in the northwestern region of Paraná State, Brazil. **Methods**: The varieties were grown in experimental plots. Crude extracts (CEs) of cones and leaves were obtained through dynamic maceration with solvent renewal (96% ethanol), followed by concentration in a rotary evaporator. Assays for sun protection factor (SPF), total phenolic content (TPC), total flavonoid content (TFC), ferric reducing antioxidant power (FRAP), 2.2-diphenyl-1-picrylhydrazyl (DPPH^•^) radical scavenging activity, and 2.2′-azinobis-(3-ethylbenzothiazoline-6-sulfonic acid) (ABTS^•^⁺) were performed to assess antioxidant activity. Chemical profiling was performed using UHPLC-MS/MS. **Results**: UHPLC-MS/MS analysis revealed the presence of phenolic and organic acids, flavonoids, phenolic aldehydes, alkaloids, and α-benzopyrone-type lactones, with high concentrations of rutin (>500 µg/g) in both cones and leaves. Total phenolic content ranged from 69.70 to 95.95 µg gallic acid equivalent/mg CE; flavonoids from 170.53 to 696.67 µg quercetin equivalent/mg CE; DPPH^•^ EC_50_ values ranged from 1.34 to 3.13 mg/mL; FRAP from 1.19 to 2.52 µM ferrous sulfate/mg; ABTS^•^⁺ from 5.11 to 22.60 mM Trolox/mg CE; and SPF ranged from 16.02 to 39.48 in the CE of *H. lupulus* cones and leaves. Conclusions: These findings demonstrate that the cultivated varieties possess antioxidant and photoprotective properties, encouraging further studies to explore their potential applications.

## 1. Introduction

*Humulus lupulus* L., commonly known as hop, is a dioecious climbing herbaceous plant belonging to the Cannabaceae family [1]. Native to temperate regions, hop cultivation typically occurs between latitudes 35° and 55° north and south of the Equator, where the world’s leading producers are concentrated, namely the United States, Germany, the Czech Republic, and the European Union [2].

Hop cultivation is mainly constrained by the plant’s photoperiodic requirements. For optimal cone yield and quality, hop plants require cold temperatures during dormancy, mild spring weather, adequate moisture from irrigation or rainfall throughout the growing season, and dry conditions during harvest [3].

In Brazil, the tropical climate and shorter photoperiods represent challenges for hop cultivation, as the country lies outside the ideal latitude range (35–55° N or S of the Equator) [4]. However, hop farming has expanded in recent years due to advances in genetic improvement and the adaptation of hop varieties to tropical conditions [5]. Currently, approximately 25 hectares are planted across various regions of Brazil, particularly in the southern states [6].

Among the hop varieties successfully adapted to tropical and subtropical Brazilian climates are Cascade, Chinook, Columbus, and Nugget [7], along with Hallertauer Mitterlfrueh, Hallertauer Magnum, Centennial, Comet, and the locally developed variety ‘Mantiqueira’. The latter was cultivated in São Bento do Sapucaí (São Paulo State), yielding two tons of Cascade hops without artificial intervention and, hence, named “Mantiqueira” [8].

However, hop production in Brazil faces several obstacles, such as an unestablished production chain and limited technology in the machinery used in various processing stages. This means that Brazil imports almost all of its raw material (cones) from the United States and Germany [9]. This has been the biggest challenge for the implementation of hops in Brazil: establishing itself as a production chain supplying the Brazilian domestic market while maintaining the same chemical and sensory characteristics as imported hops.

Globally, hop production supports multiple industries—including cosmetics, food, pharmaceuticals, and animal health—but approximately 97% is directed to the brewing industry [10,11]. The compounds found in the lupulin glands, α- and β-acids, polyphenols, and essential oils, are responsible for bitterness, aroma, and flavor, and act as natural preservatives that extend the shelf life of beer [4].

The female inflorescences of the hop plant, known as cones, are particularly valuable due to the secretion of lupulin by glandular trichomes. Lupulin is composed of resins and essential oils, with α- and β-acids (bitter acids) being its major components. The α-acids include six humulone analogs present in all hop varieties, mainly humulone, cohumulone, and adhumulone [1].

The chemical profile of hop cones includes a wide variety of phenolic compounds such as catechin, epicatechin, rutin, quercetin, and morin [12]; various phenolic acids including ferulic, protocatechuic, 4-hydroxybenzoic, vanillic, gallic, *p*-coumaric, trans-cinnamic, gentisic, caoric, and ellagic acids; and additional compounds such as syringic acid, coumarin, naringenin, and prenylflavonoids like xanthohumol [13].

Polyphenols in hops have garnered scientific interest due to their potential applications as natural additives with antimicrobial and/or antioxidant properties [14]. In vitro studies hop-derived polyphenols, they had differential beneficial effects in melanoma cells, and dermal fibroblasts [15]. In this context, the cosmetic industry has increasingly explored plant-based ingredients, especially hops, for their pharmacological properties, including antioxidant, anti-inflammatory, healing, anti-aging, moisturizing, and UV-protective activities [16], given that flavonoids, phenolic compounds, and hydroxycinnamic acid esters help protect plant tissues from UV radiation [17].

The antioxidant and photoprotective actions of phenolic compounds have highlighted *H. lupulus* as a promising natural source for multifunctional applications. Given this context and the growing interest in natural bioactive ingredients, the present study aimed to evaluate the chemical composition, antioxidant activity, and sun protection properties of cone and leaf extracts from four *H. lupulus* varieties (Cascade, Columbus, Comet, and Nugget) cultivated in the northwestern region of Paraná State, Brazil.

## 2. Results and Discussion

The definition of the solvent to be used in the preparation of the extract is a determining step in the experimental design, since it directly influences the chemical profile of the extract obtained. The choice of the extracting solvent for this study was based on Lela et al. [18], who used 96% ethyl alcohol to obtain crude extracts of *H. lupulus* varieties, including those evaluated in the present investigation, 90% ethanol due to its polarity, is a strategic choice for the extraction of hop polyphenols, a class of compounds recognized for their relevant role in the antioxidant activity of these extracts [14].

The yield of crude extracts from cones and leaves of the four varieties of *H. lupulus* is shown in Figure 1, indicating higher yields for cones when compared to the leaves.

The crude extracts of *H. lupulus* cones and leaves presented a complex composition, as evidenced by UHPLC-MS/MS analyses (Table 1 and Table 2 and Appendix A). Twenty-seven compounds were identified in the female inflorescences (cones) and 21 compounds in the leaves of the Cascade, Columbus, Comet, and Nugget varieties. The main classes detected included phenolic and organic acids, flavonoids, phenolic aldehydes, alkaloids, and α-benzopyrone-type lactones.

To further explore the bioactive profile of the crude extracts, the total phenolic and flavonoid contents were determined. The content of total phenols in the cones and leaves of *H. lupulus* of Cascade, Columbus, Comet, and Nugget varieties was investigated at concentrations of 1.0, 0.75, 0.50, and 0.25 mg/mL, as shown in Table 3. The quantification was performed separately for cones and leaves, enabling the assessment of both the influence of extract concentration and the differences between plant organs. In general, the cones exhibited higher levels of total phenols compared to the leaves. Among the cones, the Cascade, Columbus, and Nugget varieties showed the highest concentrations, whereas in the leaves, the Comet and Nugget varieties presented the highest total phenol contents.

Previous studies in the literature further support these findings. Lela et al. [18], in a study conducted at EVRA srl (Galdo di Lauria, Italy), also reported the presence of various polyphenolic compounds in hop extracts. Vergun et al. [19] analyzed different parts of the hop plant and reported elevated polyphenol levels during the flowering stage in ethanolic extracts, with values ranging from 23.76 to 54.13 mg GAE/g. Together, these results corroborate the high total phenol content observed in the present study (Table 3). The flavonoid and total phenol content was investigated by Almeida et al. [8], with hop cones of the Cascade variety grown in Brazil (Itu-SP) and in Washington (USA). The two materials were subjected to antioxidant tests, indicating Flavonoids (BR: 54.470 μg/mL; USA: 52.940 μg/mL) and in our research (193.82 μg/mL). The total phenols content was (BR: 33.930 μg/mL; USA: 27.310 μg/mL QE), and in our research (95.95 μg/mL QE). The difference found can be justified by edaphoclimatic factors, since it is the same variety.

These findings are consistent with the chemical composition data presented in Table 2 and Table 3, which revealed a wide range of phenolic compounds in both cones and leaves across all four varieties. In terms of qualitative composition, Santarelli et al. [20] identified hydroxybenzoic acids (gallic, ellagic, protocatechuic, syringic, and *p*-hydroxybenzoic acids), hydroxycinnamic acids (chlorogenic, chicoric, *p*-coumaric, ferulic, and caffeic acids), flavanols (catechin and epigallocatechin), and rutin in cones of the Cascade variety cultivated in Abruzzo, Italy compounds also identified in the present study.

In addition to total phenols, the flavonoid content was also quantified in the crude extracts of cones and leaves of *H. lupulus* from the Cascade, Columbus, Comet, and Nugget varieties at concentrations of 1.00, 0.75, 0.50, and 0.25 mg/mL, as presented in Table 4. The results revealed that among the cones, the Cascade and Columbus varieties exhibited the highest flavonoid concentrations, while among the leaves, the Comet and Nugget varieties showed the highest levels.

These findings are relevant given the structural characteristics of flavonoids and other phenolic compounds, which are directly linked to their antioxidant capacity. Analyzing the chemical structure of phenolic acids, flavonoids, and polyphenols reveals the presence of one or more hydroxyl groups conjugated to an aromatic hydrocarbon ring. This structural feature is responsible for the antioxidant activity of phenolic compounds, which may vary depending on both the number and the position of the hydroxyl groups [8]. In general, the antioxidant potential of phenolics is directly related to the quantity, number, and position of these hydroxyl groups within the molecule [21] as exemplified by the flavonoid rutin (Figure 2).

This compound was the most abundant flavonoid detected in the extracts from both cones and leaves of the four *H. lupulus* varieties studied, with concentrations exceeding 500 µg/g (Table 1 and Table 2). Its high content likely explains the elevated flavonoid levels observed in the extracts, particularly in the leaves of the Comet and Columbus varieties, which reached 696.67 and 620.27 µg quercetin equivalents per mg of crude extract, respectively (Table 4). Sotto et al. [22] also identified rutin as the major flavonoid (363 µg/g) in hydroalcoholic extracts from *H. lupulus* cones (variety not specified). Similarly, Kobus-Cisowska et al. [12], analyzing aqueous and 40% ethanolic extracts of cones from three *H. lupulus* varieties (Magnum, Lubelski, and Marynka), reported high rutin concentrations ranging from 324.4 to 1764.7 µg/g.

Of particular note are the high rutin levels found in the leaves of all four varieties investigated, with values exceeding 500 µg/g (Table 2). Keskin et al. [23] also reported higher rutin content in leaf extracts (1338.94 µg/g) than in cones (903.48 µg/g) of *H. lupulus* (variety not specified), highlighting the biotechnological potential of leaves, which are often discarded as agro-industrial waste.

The presence of rutin in both cones and leaves supports the antioxidant potential observed in the assays performed in this study. This flavonoid plays a well-documented role in neutralizing free radicals and preventing oxidative stress, in addition to downregulating inflammatory mediators such as tumor necrosis factor-alpha (TNF-α), interleukin-6 (IL-6), cyclooxygenase-2 (COX-2), and inducible nitric oxide synthase (iNOS) [24]. Furthermore, rutin has been reported to inhibit platelet aggregation and reduce capillary permeability, contributing to improved blood flow and circulation [25].

Given the richness of phenolic compounds, the antioxidant potential of the extracts was subsequently investigated using three complementary assays. The antioxidant potential of the extracts obtained from cones and leaves of the four *H. lupulus* varieties was evaluated using three methods, and the results are presented in Table 5. According to the DPPH assay, the highest antioxidant activities were observed in the cone extracts of the Cascade and Columbus varieties and the leaf extract of the Comet variety. In the FRAP assay, the cones of the Comet and Nugget varieties, along with the leaves of the Columbus variety, exhibited the highest antioxidant potential. Notably, the ABTS assay revealed a distinct pattern in which the leaf extracts demonstrated superior antioxidant activity compared to the cones, with particular emphasis on the leaves of the Cascade variety (Table 5).

It is important to note that no single method can provide a definitive measure of antioxidant capacity, as each technique evaluates different mechanisms of action. For this reason, we adopted multiple assays to obtain a more comprehensive assessment of antioxidant potential. As emphasized by Alam et al. [26], the results of antioxidant evaluations may vary depending on the method used, the chemical nature of the sample, and the interaction of compounds within the extract. Therefore, the combined use of DPPH, FRAP, and ABTS assays enhances the reliability and interpretative depth of the antioxidant profile [27].

No less important than the antioxidant protocols used is the influence of genetic variability associated with climatic factors such as sun exposure, temperature, and water availability on the chemical composition of *H. lupulus* cultures [11]. In the present study, the impact of these variables on the antioxidant response was observed, as shown in Table 5, indicating a significant difference in the antioxidant response of the four varieties investigated. For optimal growth, hop plants need particular climatic conditions such as exposure to low temperatures during dormancy, mild temperatures in spring, sufficient moisture from irrigation or rainfall throughout the season, and dry weather during harvest time [3]. In this sense, Forster et al. [28] evaluated the impact of climatic conditions on the biogenesis of various compounds in hops in European hop-producing countries. The authors assert that high temperatures and low moisture content are the factors that most negatively impact hop production and its metabolites. They report in their studies that the compounds most sensitive to these parameters are α-acids, with a 30% decrease in production, followed by a 17% decrease in essential oil synthesis and phenolic compounds, with a 9% sensitivity level.

Therefore, we consider that the total phenol and flavonoid contents found in the present experiment (Table 3 and Table 4) may also have been affected by high temperatures (27.09 ± 4.00 °C) from November 2022 to January 2023, as well as low humidity reflected by the low rainfall (5184 ± 10.04 mm). Another factor that may have affected the chemical composition and interfered with the antioxidant response is the soil where the hops are grown: sandy texture, low water retention capacity, and low organic matter content [29]. According to Sposito et al. [30], hop plants grow well in different types of soil, as long as they are fertile and able to retain moisture.

*p*-Hydroxybenzoic acid, also identified in high concentrations in both cones and leaves across all four *H. lupulus* varieties (Table 1 and Table 2), is considered the fundamental monohydroxybenzoic structure shared by a wide range of phenolic acid derivatives. While monophenols generally exhibit lower radical scavenging efficiency compared to polyphenols, structural variations, particularly the introduction of electron-donating or electron-withdrawing groups at specific positions on the aromatic ring, can significantly enhance the antioxidant potency of the resulting compounds. Functional groups substituted at the ortho or para-positions of the phenolic ring are more effective than those at the meta-position in modulating antioxidant performance. Among the most well-known derivatives of *p*-hydroxybenzoic acid containing such functional groups are protocatechuic acid (3.4-dihydroxybenzoic acid), gallic acid (3.4.5-trihydroxybenzoic acid), vanillic acid (3-methoxy-4-hydroxybenzoic acid), and syringic acid (3.5-dimethoxy-4-hydroxybenzoic acid) (Figure 3) [31].

The primary mechanism of action of phenolic antioxidants involves the neutralization of stable radical species, as detected by the ABTS assay [21] and the DPPH assay [32], as well as the reduction in ferric ions, assessed by the FRAP method [33]. These antioxidant assays were employed in the present study, with results detailed in Table 5, showing that the cones showed a greater antioxidant response than the leaves in the three assays.

The DPPH, FRAP, and ABTS assays revealed higher antioxidant potential in cones compared to leaves (Table 5). Hop extracts were also analyzed by Lyu et al. [34] who used DPPH and ABTS assays to investigate nine varieties: Saaz (from four different growing regions), Calypso, Cascade, Cluster, El Dorado, and Magnum grown in South Korea. These authors found that the Saaz 3 and Colorado varieties had greater potential using the DPPH (IC_50_ = 0.12 mg/mL) and ABTS (0.19 mmol Trolox/g dw) methods.

Arruda et al. [35] investigated the antioxidant action by the ABTS method of cone extracts obtained by maceration of nine varieties of *H. lupulus* grown in Nova Friburgo, Rio de Janeiro, and Serra da Mantiqueira, São Paulo, Brazil. Among these, the varieties Cascade and Columbus have IC_50_ values of 0.079 mMol and 0.139 mMol Trolox/g dw, respectively. In Brazil, these two locations are favorable for hop cultivation due to their climate and altitude, contributing to their excellent antioxidant response.

Corroborating these findings, Biskup et al. [21] investigated the antioxidant potential (IC50) of 10 water-soluble phenolic compounds using the ABTS and FRAP methods. The authors observed that phenols with a greater number of hydroxyl groups had greater antioxidant capacity, as indicated by lower IC_50_ values, 4.33 mM and 5.23 mM, respectively, for gallic acid and pyrogallol. Another finding in this experiment was that the type of spacer between the carboxylic acid and the benzene ring influences its activity, which increases for the methylene group in phenolic acids. They also found that phenols with two hydroxyl groups attached to the aromatic ring in the ortho position suppress the ABTS^•+^ radical more strongly than compounds with hydroxyl groups substituted in the meta position. In the present study, protocatechuic acid contains two hydroxyl groups directly attached to the aromatic ring, one in the ortho position and the other in the para position in relation to the carboxylic group (Figure 3). This configuration may have contributed to the antioxidant response observed in both leaf and cone extracts in the ABTS assay (Table 5) since protocatechuic acid was identified in considerable amounts in cone extracts from the Nugget (416.96 µg/g), Columbus (335.85 µg/g), Cascade (272.16 µg/g), and Comet (211.05 µg/g) varieties (Table 1) and was also detected in the leaves of all four varieties, with a prominent concentration in the Columbus variety (440.34 µg/g) (Table 2).

Protocatechuic acid is also recognized for its ability to scavenge peroxyl radicals in polar aqueous environments and to act as an antiradical protector in the apolar context of lipid-based systems. It is capable of attenuating oxidative stress by enhancing the activity of antioxidant enzymes such as glutathione peroxidase (GSH-PX) and superoxide dismutase (SOD), while reducing the activity of xanthine oxidase (XOD) and NADPH oxidase (NOX), as well as decreasing malondialdehyde (MDA) levels [36].

The occurrence of high concentrations of protocatechuic acid in *H. lupulus* has also been reported in the literature. Almeida et al. [8] identified this phenolic acid in the hydroethanolic extract of Cascade hop cones (10.0 µg/g) using ultrasound-assisted extraction. Similarly, Keskin et al. [23] detected protocatechuic acid in methanolic extracts of *H. lupulus* cones (5.71 µg/g), obtained using a 2-h sonication protocol with an ultrasonicator.

Protocatechuic acid is a water-soluble benzoic acid derivative with well-documented biological activities, including antioxidant and anti-inflammatory effects [37], as well as anti-atherosclerotic, antineoplastic, analgesic, antibacterial, hepatoprotective, antiviral, and hepatorenal protective properties [36,38].

The presence of a carbonyl group in the form of an ester also contributes to increased antioxidant activity, as observed in chlorogenic acid (Figure 4), which was detected in high concentrations in both cones and leaves of *H. lupulus*. This phenolic acid is the primary representative of hydroxycinnamic acids and is chemically classified as an ester formed between caffeic acid and quinic acid. Its molecular structure features vicinal hydroxyl groups on an aromatic ring, which are associated with in vitro anti-mutagenic, anti-carcinogenic, and antioxidant activities, particularly in the scavenging of reactive oxygen species (ROS) [39].

Chlorogenic acid has also been reported to exert anti-inflammatory, anti-apoptotic, and antihypertensive effects [39,40]. Furthermore, it influences lipid and glucose metabolism in various genetic and metabolic disorders. It has been shown to reduce plasma concentrations of high-density lipoprotein (HDL), total cholesterol, and low-density lipoprotein (LDL) in rat models [41]. In addition, several studies have highlighted its therapeutic potential, particularly in the prevention or treatment of fibrosis and cancer [42,43,44,45].

*p*-Coumaric acid (4-hydroxycinnamic acid) (Figure 5), another representative of the hydroxycinnamic acid class, was detected in high concentrations in the cone extracts of the Comet (202.33 µg/g) and Nugget (176.68 µg/g) varieties. In the leaf extracts, higher levels were found in the Nugget (351.902 µg/g) and Columbus (324.489 µg/g) varieties. The presence of a hydroxyl group in the para position (Figure 4) may have contributed to the antioxidant activity observed in this study. What is supported by findings from Kiliç and Yesiloglu [46], who reported that *p*-coumaric acid exhibits chemoprotective properties by scavenging carcinogenic nitrosating agents in various biological compartments, including salivary and gastric fluids. Hydroxybenzaldehyde, a phenolic compound, was found in high concentrations in the cone extracts of the Comet (183.84 µg/g), Nugget (143.94 µg/g), and Cascade (116.78 µg/g) varieties. It was also detected in the leaf extracts of Comet (119.836 µg/g), Columbus (117.360 µg/g), and Nugget (109.823 µg/g), with lower levels observed in Cascade leaves (33.898 µg/g). Considered a component of the plant’s defense system and a response to external stress, the bioactivity of secondary metabolites such as polyphenols and saponins depends on factors including chemical structure and concentration [47,48].

In addition to the antioxidant capacity, the extracts were also tested for their photoprotective potential. The extracts obtained from the cones and leaves of the four *H. lupulus* varieties were evaluated for their sun protection factor (SPF), with results presented in Table 6. Cone extracts exhibited higher SPF values (33.27 to 39.48) compared to leaf extracts (16.02 to 22.37), both at a concentration of 1.0 mg/mL. According to Brazilian regulatory standards, RDC No. 629. ANVISA. [49], SPF values ≥ 6.0 are required for consideration in sunscreen development. Based on these criteria, the results suggest that extracts from both plant organs across all varieties possess promising photoprotective potential.

In addition to SPF, other photoprotective parameters were assessed, including the UVA/UVB ratio, anti-UVA protection factor, and critical wavelength (λc), as shown in Table 7. For the tested concentrations (1.0 to 0.25 mg/mL), anti-UVA protection values ranged from 1.62 to 2.02 for cone extracts and from 2.10 to 3.03 for leaf extracts. These values correspond to the “ultra” protection category, based on the classification scale proposed by Boots [50] and Wang et al. [51], which includes the levels: very low, moderate, good, superior, maximum, and ultra. Supporting these findings, Kurzawa et al. [17] reported an “ultra” level of UVA protection (value of 5.0) for a water–glycolic hop cone extract tested at lower concentrations (0.1–0.4 mg/mL).

These photoprotective properties may be attributed, at least in part, to the flavonoid composition of the extracts. Flavonoids are among the most extensively studied phytochemicals in the context of photoprotection, as they help protect plants from UV radiation and mitigate UV-induced reactive oxygen species (ROS). As such, flavonoids exert three significant photoprotective actions: (i) absorption of UV radiation, (ii) direct and indirect antioxidant activity, and (iii) modulation of signaling pathways involved in oxidative stress and cellular response [52].

The ultraviolet absorption spectrum of flavonoids typically shows two absorption peaks: one between 240 and 280 nm (UVB) and another between 300 and 550 nm (UVA), making them suitable candidates for natural sunscreen applications [53]. Structurally, flavonoids share a common backbone that can be classified into subgroups, such as isoflavones, neoflavonoids, among others, based on the position of the B-ring on the C-ring and the degree of oxidation and unsaturation (Figure 6). The presence of chromophoric groups in their structure contributes to their ability to absorb ultraviolet radiation [54].

High concentrations of flavonoids, such as rutin, in plants have also been shown to prevent UV-induced generation of reactive oxygen species (ROS) [55]. In this context, the presence of rutin, along with other flavonoids such as catechin, epicatechin, quercetin, morin, and naringenin, identified in the cone and leaf extracts of all four *H. lupulus* varieties (Table 1 and Table 2), may have contributed to the high SPF values observed in the cone extracts (Table 1 and Table 6; Figure 5).

The flavonoid morin was identified exclusively in cone extracts of the Nugget (265.37 µg/g), Cascade (150.76 µg/g), and Comet (141.04 µg/g) varieties, and lower amounts in the Columbus variety (86.94 µg/g) (Table 1). Morin has been studied for its role in the prevention and treatment of chronic disorders associated with oxidative stress and inflammation. Sharma et al. [56] reported that morin significantly attenuates the activation of the NF-κB signaling pathway and downregulates the expression of its inflammatory mediators, including tumor necrosis factor-alpha (TNF-α), interleukin-6 (IL-6), and prostaglandin E2 (PGE-2), in rats treated with a carcinogen. In a related study, Yong and Ahn [57] demonstrated the applicability of morin as a skin-protective antioxidant, showing cytoprotective activity against UVA-induced free radicals. The authors observed that UVA exposure led to a concentration-dependent increase in the expression of antioxidant enzymes such as glutathione peroxidase (GPx), catalase (CAT), heme oxygenase (HO), and nuclear factor erythroid 2–related factor 2 (NRF2), supporting the role of morin in eliminating oxidative species and offering primary cytoprotection against acute oxidative stress.

Quercetin was identified in cone extracts of all four hop varieties, with concentrations ranging from 101.97 to 164.32 µg/g (Table 1). This flavonoid is recognized for its potent antioxidant activity, mediated by multiple mechanisms including reactive oxygen species scavenging, inhibition of lipid peroxidation, and metal ion chelation [58]. Among flavonoids, quercetin exhibits one of the strongest radical-scavenging abilities against hydroxyl, peroxyl, and superoxide anion radicals. Its antioxidant properties are attributed to three key structural features: (i) the 3′.4′-dihydroxycatechol configuration in the B-ring, which enables the formation of more stable phenoxyl radicals after hydrogen donation; (ii) the presence of a 2.3-double bond in conjugation with a 4-carbonyl group in the C-ring, which facilitates electron delocalization from the B-ring to the C-ring; and (iii) the 3-hydroxyl group in combination with the 2.3-double bond, which enhances resonance stabilization and promotes electron delocalization throughout the molecule (Figure 5) [59]. Among the previously reported effects of quercetin are the inhibition of cancer cell proliferation, reduction in blood pressure, and cholesterol levels. Additionally, studies indicate a strong potential in the treatment of cardiovascular diseases [60].

The flavonoid catechin was one of the major compounds identified in the cone extracts of the Nugget variety (177.18 µg/g), and was found in lower amounts in the Cascade (54.83 µg/g), Comet (53.42 µg/g), and Columbus (51.30 µg/g) varieties. In the leaf extracts, catechin was detected in the Comet (74.206 µg/g) and Nugget (71.339 µg/g) varieties, but was absent in the leaves of Cascade and Columbus.

Hop plants are almost exclusively cultivated for the brewing industry, as the resins and essential oils from the female cones are used to impart flavor to beer [61]. However, studies on the four hop varieties have shown that they also contain flavonoids and phenolic aldehydes in both the cones and leaves (Table 1 and Table 2), indicating potential industrial applications beyond beer production.

In cosmetics, flavonoids are valued for their sunscreen, anti-aging, and anti-inflammatory properties. In the food industry, these compounds help extend the shelf life of foods thanks to their antimicrobial and antioxidant properties [62]. As for phenolic aldehydes, such as dehydroxybenzaldehydes in ortho, para, and meta positions (o-, p-, and m-hydroxybenzaldehydes), they are of great importance both in biology and in industry, as they participate in proton transfer discussions, essential in processes ranging from acid–base reactions to enzyme-catalyzed solutions [63]. Finally, it is worth highlighting vanillin, a phenolic aldehyde widely used as a flavoring in food and cosmetics, due to its sensory and functional properties [64].

Although the extracts demonstrated significant levels of bioactive polyphenols, the present study did not assess their chemical stability during storage or processing. Phenolic compounds are known to be sensitive to pH, light, temperature, and oxygen exposure [65], which can influence their bioactivity and shelf life. Therefore, future studies should investigate the stability of these compounds in various delivery systems and under different storage conditions to ensure efficacy and formulation compatibility in prospective therapeutic applications.

The results obtained in this study demonstrate that both the cones and leaves of the four *H. lupulus* varieties tested exhibit significant antioxidant and photoprotective properties, attributed to their high content of identified phenolic compounds. The leaves, often regarded as agro-industrial waste, can be repurposed and valorized in multiple ways, suggesting their potential use as antioxidant ingredients in the food, pharmaceutical, and cosmetic industries. These findings support further investigations aimed at expanding the understanding of their bioactive potential and formulation viability.

## 3. Materials and Methods

### 3.1. Plant Material and Field Conditions

The experiment was conducted in an experimental area belonging to the State University of Maringá (UEM), at the Umuarama Regional Campus (Paraná State), located at 23°47′25′′ South latitude and 53°15′31′′ West longitude. The soil in which the experiment was set up is derived from the Caiuá Sandstone soil and is classified as a dystrophic red latosol, which can be argisols and neosols to a lesser extent [29].

The crop began to be planted in November 2022, and four hop varieties were used, namely Columbus, Cascade, Comet, and Nugget, with a total of 204 seedlings arranged in an experimental treatment.

Botanical identification was carried out, and specimens were deposited in the Herbarium of the Universidade Paranaense under number 340. The species was documented in the National System of Genetic Heritage Management and Associated Traditional Knowledge (SisGen, as abbreviated in Portuguese) with identification number AC98839. The varieties were cultivated as climbers, following management recommendations. Fertilization was carried out according to the soil samples collected, observing the acidity correction according to the saturation obtained in the chemical analyses [30]. After three months (January 2023), the cones and leaves used to prepare the crude extracts were collected for the first time. The temperature and rainfall index during the growing period were 27.09 ± 4.00 °C and 5.184 ± 10.04 mm, respectively.

### 3.2. Extract Obtention

The female inflorescences (cones) and leaves from the four varieties were dried at room temperature. In the Columbus variety 110 g were obtained for the cones, 50 g for the leaves, in the Cascade variety 85 g for the cones and 102 g for the leaves, in the Comet variety, 47 g of cones and 116 g of leaves were used and in the Nugget variety 43 g for the cones and 100 g for the leaves. Afterwards, the leaves and cones were pulverized in an industrial blender model until they reached a particle size of 850 mm. The powder obtained was subjected to a dynamic maceration process with the solvent renewed using ethyl alcohol (96 °C) until the plant material was exhausted [18,66]. The filtrate was then concentrated under reduced pressure in a rotary evaporator (Tecnal TE-211 model, Piracicaba, São Paulo, Brazil) at 40 °C until the CE was obtained. The yield (%) of the extracts was calculated from the ratio of the dry mass of the plant material (g) divided by the mass of the crude extract (g).

### 3.3. Chemical Identification of the Crude Extracts of H. Lupulus Cones and Leaves by Ultra-Performance Liquid Chromatography Coupled with High-Resolution Mass Spectrometry (UHPLC-MS/MS)

The crude extract of *H. lupulus* cones and leaves was analyzed using ultra-performance liquid chromatography, coupled with tandem mass spectrometry (UHPLC-MS/MS) (Shimadzu^®^ model 8050 MS and Nexera^®^ X2 HPLC, Kyoto, Japan), equipped with an electrospray ionization source (EIS). EIS was used in negative and positive mode, scanned with multiple reaction monitoring (MRM) mode. Collision energy was −15 V (positive) and 30 V (negative). An interface voltage of 3 kV, current of 7µA, temperature of 300 °C, and flow rates of nebulizing gas and drying gas were 3 L/min and 10 L/min, respectively. Argon was used as a collision gas at a maximum pressure of 20 mPa. Chromatographic separation was carried out using a C18 column (5 µm, 150 mm × 4.6 mm Shimadzu^®^). The mobile phases were Milli-Q water acidified with 0.1% formic acid (A) and methanol-grade MS (Merck^®^ (Darmstadt, Germany)) (B). They were operated in linear gradient mode, 1–9 min (20% B), 10–15 min (40% B), and 16–30 min (10% B), at a flow rate of 0.5 mL/min, at 40 °C. The samples were filtered (PVDF hydrophobic membrane, pore size 0.45 μm, and 25 mm diameter), and the injection volume was 1 µL. Analytical curves (10–250 µg/L) were obtained for the following compounds: catechol, morin, isovanillin, gallic acid, quercetin, hydroxybenzaldehyde, naringenin, syringaldehyde, chlorogenic acid, syringic acid, protocatechuic acid, vanillic acid, salicylic acid, vanillin, ferulic acid, *p*-hydroxybenzoic acid, naringin, *p*-coumaric acid, caffeic acid, coniferyl aldehyde, sinapic acid, syringaldazine, catechin, sinapaldehyde, luteolin, rutin, theobromine, epicatechin, baicalin, chrysin, quinic acid, malic acid, kaempferol, coumarin, caffeine, resorcylic acid, nicotinic acid, and fumaric acids. The equations obtained for each analytical curve and their determination coefficients are shown in Appendix A.

### 3.4. Evaluation of the Total Phenol Content (TP) of the Crude Extracts of H. Lupulus Cones and Leaves

The content of total phenols (TPs) present in the CEs of the leaves and cones of *H. lupulus* was determined using spectroscopy in the visible region using the Folin–Ciocalteu method according to Swain and Hillis [67] with modifications by Sá [68]. The extract samples were diluted in methanol at a concentration of 1.0 mg/mL. The reagent solution consisted of 155 μL of Folin–Ciocalteu solution and 125 μL of sodium carbonate solution, followed by 20 μL of the diluted sample (1 mg/mL) in each well of the microplate.

The mixture was left to stand in the absence of light for 60 min and read using the SpectraMax Plus384 Microplate Reader at 760 nm, in triplicate. The calibration curve was obtained using seven dilutions of gallic acid (0–100 µg/mL). The equation of the calibration curve obtained by linear regression is Equation (1):A = 0.0196 C–0.031   (R^2^= 0.9997)(1)
where A represents the measured absorbance, C the concentration of gallic acid equivalents, and R^2^ represents the coefficient of determination for the multiple regression. The results were expressed as µg of gallic acid equivalent (GAE) per mg of sample.

### 3.5. Evaluation of the Flavonoid Content of the Crude Extracts of H. Lupulus Cones and Leaves

The total flavonoid content was determined using the aluminum chloride colorimetric method according to Alves and Kubota [69]. The CEs from the leaves and cones of *H. lupulus* were solubilized in methanol to obtain concentrations of 1.00, 0.75, 0.50, and 0.25 mg/mL. An aliquot of 0.5 mL of extract was mixed with 0.5 mL of 2% (*w*/*v*) aluminum chloride solution in methanol and kept at room temperature for 10 minutes. The change in absorbance was determined at 425 nm. The aluminum chloride solution was used as an analytical control. The concentration of flavonoids was calculated according to the quercetin standard curve (5–40 µg/mL). The results were expressed in µg quercetin equivalents (EQ) per mg of extract.

### 3.6. Antioxidant Activity

#### 3.6.1. Scavenging of 2.2-Diphenyl-1-Picrylhydrazyl (DPPH•) Free Radicals

The methodology described by Rufino et al. [70] was used to determine the free radical scavenging capacity of DPPH. An aliquot of 10 µL of the different concentrations of the crude extracts of the leaves and cones of *H. lupulus* at concentrations (1.0, 0.75, 0.5, and 0.25 mg/mL) and 290 µL of methanolic DPPH^•^ solution (60 µM) were added to a SpectraMax Plus384 Microplate Reader and kept for 30 minutes, and the absorbance was measured at 515 nm. For the negative control, 10 µL of methanol was used in the DPPH solution (60 µM). The total antioxidant capacity of the extracts and fractions was calculated using a standard solution of quercetin (60 µM) as a 100% reference. Based on the correlation between absorbance and concentration of the antioxidant sample, the concentration needed to reduce 50% of the free radicals (IC50) was determined.

#### 3.6.2. Ferric Reducing Antioxidant Power (FRAP)

The preparation of the FRAP reagent followed the methodology of Benzie and Strain [71], modified by Rufino et al. [72], of which 25 mL of acetate buffer (0.3 M), 2.5 mL of aqueous solution of 2.4.6-Tris (2-pyridyl)-striazine (TPTZ—10 mM), 2.5 mL of aqueous solution of ferric chloride (20 mM) and 3 mL of distilled water were combined. The reagent solution consisted of 10 μL of the CEs from the leaves and cones of *H. lupulus* at concentrations of 1.00, 0.75, 0.50, and 0.25 mg/mL and 290 μL of the FRAP reagent in each well of the microplate. The mixture was placed in the SpectraMax Plus384 Microplate Reader and kept at 37 °C for 30 minutes. Absorbance was read at 595 nm. The percentage of antioxidant activity was calculated using a standard curve of ferrous sulphate (0–2000 μM). The antioxidant activity was expressed as μM ferrous sulphate/mg of sample.

#### 3.6.3. ABTS•⁺ Radical Method (2.2′azinobis-(3ethylbenzthiazoline-6-Sulfonic Acid))

To determine antioxidant activity using the ABTS^•^⁺ radical method, the methodology described by Rufino et al. [73] was used. Initially, the ABTS^•^⁺ radical was formed from the reaction of 5.0 mL of 7 mM ABTS with 88 μL of 140 mM potassium persulfate, which were incubated at room temperature and without light for 16 hours. After this time, 1 mL of the solution was diluted in ethanol until a solution with an absorbance of 0.70 nm ± 0.05 nm at 734 nm was obtained. Microplates with 96 wells were used to carry out the analysis, with 10 μL of the extracts (at concentrations of 1.0, 0.75, 0.50, and 0.25 mg/mL) and 290 μL of the solution containing the ABTS^+^ radical were added to each well. The mixture was placed in the SpectraMax Plus384 Microplate Reader, and the absorbance was determined after 30 minutes of reaction. The synthetic antioxidant Trolox was used as a standard solution at concentrations of 100, 500, 1000, 1500, and 2000 μM in ethanol. All the readings were carried out in triplicate, and the results were expressed as mM of Trolox per gram of extract.

#### 3.6.4. In Vitro Determination of Sun Protection Factor (SPF)

The extracts of leaves and cones obtained from the four plant varieties were assessed for their in vitro sun protection factor (SPF), following a previously described methodology with modifications [74]. The extracts were redissolved in ethanol to prepare solutions at 1.00, 0.75, 0.50, and 0.25 mg/mL concentrations. Absorbance measurements were performed in triplicate using a SpectraMax Plus 384 microplate reader (Molecular Devices, San Jose, CA, USA) across the wavelength range of 290 to 320 nm, with 5 nm intervals. Ethanol served as the analytical blank.

The SPF values were calculated according to Equation (2).SPF = CF Σ EE(λ)I(λ)Abs(λ)(2)
where CF = 10 (correction factor); EE(λ) = erythemal effect spectrum; I(λ) = solar intensity spectrum; Abs(λ) = absorbance of the sample at wavelength λ; the EE(λ) x I(λ) values are constant [75].

### 3.7. Determination of the UVA/UVB Ratio and Critical Wavelength (λc)

The UVA/UVB ratio corresponds to the proportion between the area under the absorption curve in the UVA region (320–400 nm) and that in the UVB region (290–320 nm), as described by Wu et al. [76]. Ratios closer to 1.0 indicate a higher level of UVA protection.

To determine this ratio, the extracts were redissolved in ethanol at concentrations of 1.00, 0.75, 0.50, and 0.25 mg/mL. Absorbance spectra for each dilution were recorded in triplicate from 290 to 400 nm, at 5 nm intervals, using a SpectraMax Plus 384 microplate reader (Molecular Devices, San Jose, USA). Ethanol was used as the analytical blank. The UVA/UVB ratios were computed based on Equation (3):(3)$$UVA/UVB=\frac{\int_{320}^{400}A\left(\lambda \right)d\lambda \int_{320}^{400}d\lambda }{\int_{290}^{320}A\left(\lambda \right)d\lambda \int_{290}^{320}d\lambda }$$
where A(λ) = mean absorbance at each wavelength; d(λ) = wavelength interval between measurements; the photoprotective performance of the extracts in the UVA region was further classified according to Boot’s star rating system (Table 8) [50].

In addition, the absorption spectra were used to determine the critical wavelength (λc), defined as the wavelength below which 90% of the absorbance area is located. A critical wavelength equal to or greater than 370 nm is considered indicative of broad-spectrum UV protection [51].

## 4. Statistical Analysis

All analyses were carried out in triplicate. The data were subjected to analysis of variance (ANOVA). Data normality was assessed using the Shapiro–Wilk test and the Bartlett test to assess homogeneity. The data were compared using the Tukey test (*p* ≤ 0.05). SPSS Statistics 22 software, StatSoft Statistics 10.0, South America, 2022, was used for the analysis.

## 5. Conclusions

These findings indicate that even with Brazil’s challenging climate aspects, the Cascade, Columbus, Comet, and Nugget varieties of *Humulus lupulus* grown in the northwestern region of Paraná State, Brazil, possess a rich and well-balanced chemical composition, including compounds such as phenolic and organic acids, flavonoids, phenolic aldehydes, alkaloids, and α-benzopyrone lactones. Of note is the presence of rutin (>500 µg/g) in the cones and leaves of all four varieties. Furthermore, the extracts demonstrated antioxidant and photoprotective potential, which opens the door to new applications in sectors such as cosmetics, food, and even sun protection. These results encourage continued research to explore the potential of these varieties further, promoting the sustainable and innovative use of hops in Brazil.

## Figures and Tables

**Figure 1 pharmaceuticals-18-01229-f001:**
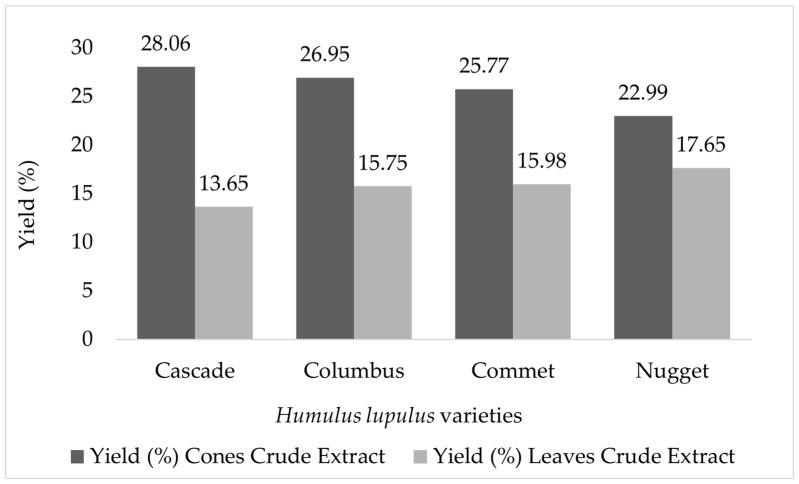
Yield (%) of crude extract from cones and leaves of *Humulus lupulus* varieties Cascade, Columbus, Comet, and Nugget.

**Figure 2 pharmaceuticals-18-01229-f002:**
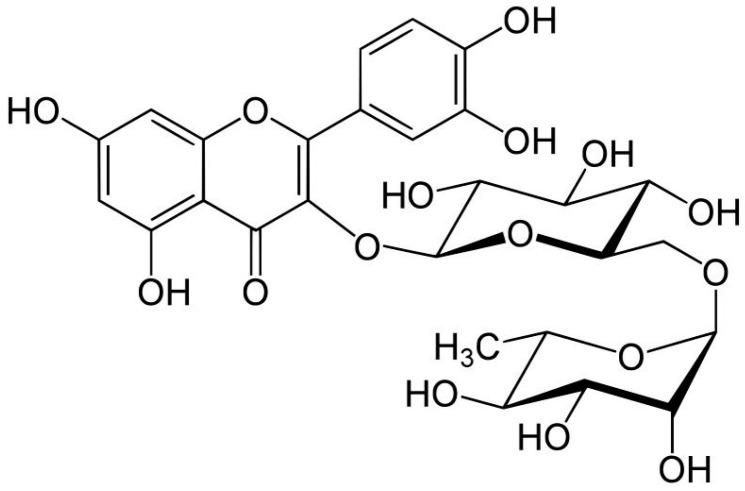
Rutin structure.

**Figure 3 pharmaceuticals-18-01229-f003:**
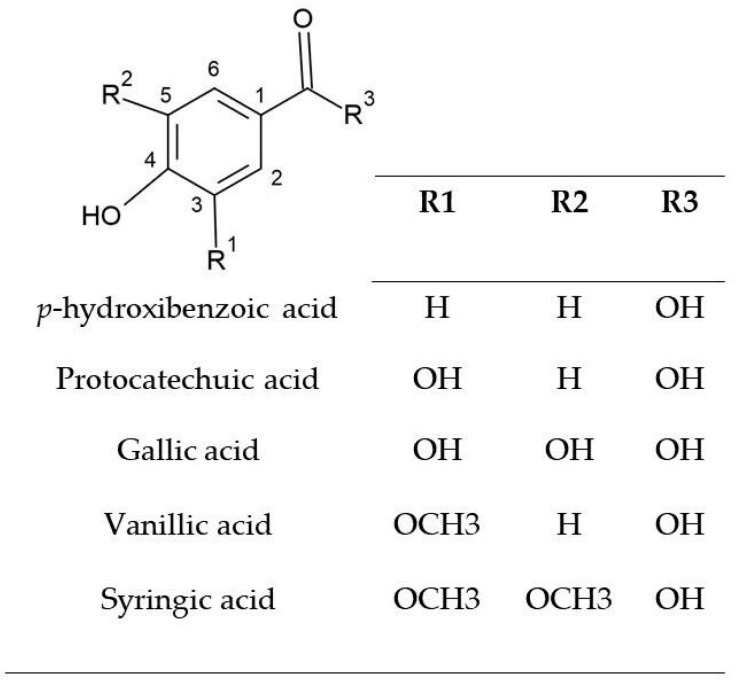
Molecular structure of *p*-hydroxybenzoic acid and its derivatives.

**Figure 4 pharmaceuticals-18-01229-f004:**
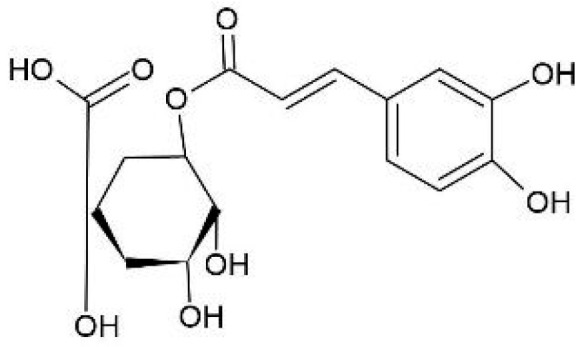
Chlorogenic acid.

**Figure 5 pharmaceuticals-18-01229-f005:**
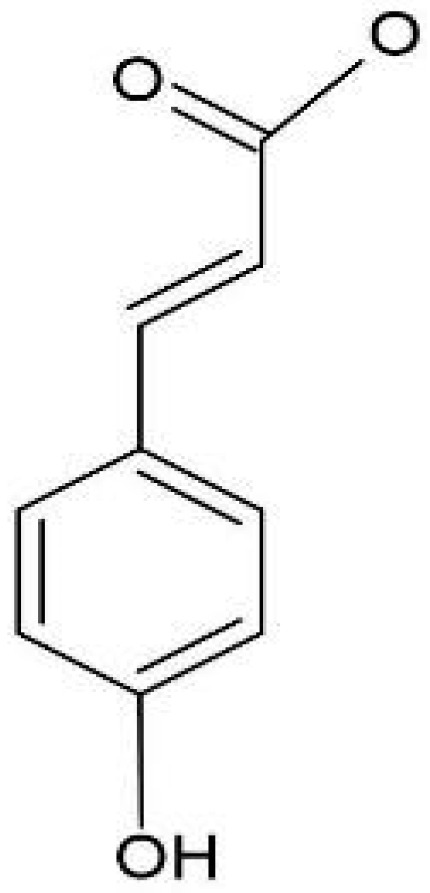
*p*-coumaric acid.

**Figure 6 pharmaceuticals-18-01229-f006:**
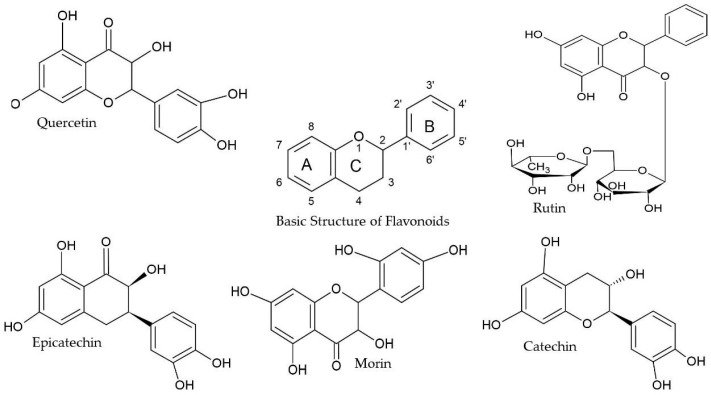
Flavonoids identified in *Humulus lupulus* cones and leaves from Cascade, Columbus, Comet, and Nugget varieties.

**Table 1 pharmaceuticals-18-01229-t001:** Chemical composition of *Humulus lupulus* L. cones crude extract (CE) from Cascade, Columbus, Comet, and Nugget varieties (µg/g).

Compound	Precursor and Productions *m*/*z*	Cascade	Columbus	Comet	Nugget
	[M-H]				
	Phenolic Acid				
Chlorogenic acid	354.80 > 162.95	107. 21 ± 5.42 ^B^	185.41 ±15.80 ^A^	59.12 ± 1.34 ^C^	127.99 ± 2.17 ^B^
Sinapic acid	225.00 > 206.95	66.45 ± 0.58 ^B^	37.28 ± 0.41 ^C^	31.40 ± 0.45 ^D^	118.62 ± 1.60 ^A^
Gallic acid	169.20 > 124.90	92.62 ± 3.75 ^B^	106.38 ± 1.89 ^A^	32.43 ± 1.45 ^D^	76.81 ± 2.15 ^C^
Protocatechuic acid	153.20 > 108.95	272.16 ± 15.78 ^BC^	335.85 ± 20.23 ^B^	211.05 ± 9.62 ^C^	416.96 ± 23.06 ^A^
Syringic acid	197.20 > 182.05	12.35 ± 0.29 ^BC^	15.63 ± 1.29 ^B^	10.29 ± 0.77 ^C^	23.22 ± 1.55 ^A^
Vanillic acid	167.40 > 152.00	59.59 ± 4.30 ^A^	35.78 ± 3.07 ^B^	35.17 ± 0.74 ^B^	63.44 ± 0.94 ^A^
*p*-Hydroxybenzoic acid	137.20 > 92.90	226.60 ± 7.66 ^B^	129.27 ± 1.71 ^D^	175.01 ± 10.26 ^C^	330.61 ± 14.33 ^A^
*p*-Coumaric acid	163.20 > 119.00	109.67 ± 5.55 ^B^	85.87 ± 3.41 ^B^	202.33 ± 11.66 ^A^	176.68 ± 2.28 ^A^
Caffeic acid	179.20 > 135.00	14.08 ± 1.13 ^B^	17.34 ± 1.40 ^B^	13.58 ± 1.26 ^B^	27.26 ± 0.64 ^A^
Ferulic acid	193.20 > 133.95	122.55 ± 7.80 ^B^	92.77 ± 2.15 ^C^	120.25 ± 2.27 ^B^	143.96 ± 0.04 ^A^
Quinic acid	191.20 > 84.80	64.25 ± 1.51 ^C^	125.29 ± 4.21 ^A^	104.75 ± 3.12 ^B^	61.99 ± 0.80 ^C^
	Organic Acids				
Malic acid	133.20 > 114.90	191.73 ± 0.14 ^D^	223.21 ± 8.87 ^C^	354.37 ± 10.89 ^A^	309.66 ± 2.55 ^B^
Fumaric acid	115.20 > 71.00	193.09 ± 3.64 ^B^	120.08 ± 8.85 ^C^	82.05 ± 7.82 ^D^	193.09 ± 3.64 ^A^
Nicotinic acid	122.40 > 77.90	46.52 ± 1.95 ^A^	19.98 ± 1.59 ^C^	14.56 ± 0.19 ^C^	37.84 ± 2.82 ^B^
	Flavonoids				
Catechin	289.10 > 245.10	54.83 ± 0.17 ^B^	51.30 ± 4.85 ^B^	53.42 ± 4.94 ^B^	177.18 ± 0.66 ^A^
Morin	302.80 > 152.90	150.76 ± 9.48 ^B^	86.94 ± 2.78 ^C^	141.04 ± 1.58 ^B^	265.37 ± 10.28 ^A^
Epicatechin	289.10 > 245.10	53.38 ± 4.89 ^B^	56.32 ± 1.15 ^B^	71.29 ± 6.68 ^B^	174.77 ± 13.28 ^A^
Quercetin	301.10 > 151.00	117.35 ± 11.65 ^B^	101.97 ± 7.70 ^B^	159.57 ± 0.01 ^A^	164.32 ± 12.59 ^A^
Rutin	609.00 > 300.00	>500 µg/g ^A^	>500 µg/g ^A^	>500 µg/g ^A^	>500 µg/g ^A^
Luteolin	286.90 > 152.85	<1 µg/g ^A^	<1 µg/g ^A^	<1 µg/g ^A^	<1 µg/g ^A^
Naringenin	271.10 > 151.00	13.19 ± 1.18 ^B^	14.42 ± 0.33 ^B^	4.44 ± 0.10 ^C^	35.77 ± 2.57 ^A^
	Phenolic Aldehydes				
Hydroxybenzaldehyde	123.10 > 76.9500	116.78 ± 7.10 ^B^	77.74 ± 5.29 ^C^	183.84 ± 9.81 ^A^	143.94 ± 6.33 ^B^
Vanillin	151.40 > 136.05	101.14 ± 1.64 ^A^	45.22 ± 2.15 ^B^	21.98 ± 1.47 ^C^	92.16 ± 3.75 ^A^
Isovaniline	153.10 > 65.00	21.21 ± 0.07 ^B^	11.21 ± 0.15 ^C^	11.72 ± 0.94 ^C^	38.48 ± 0.34 ^A^
Syringaldehyde	183.00 > 94.95	27.75 ± 0.56 ^A^	11.20 ± 0.82 ^C^	16.10 ± 0.10 ^B^	27.06 ± 0.26 ^A^
	Alkaloids				
Caffeine	194.90 > 137.95	6.71 ± 0.28 ^A^	3.55 ± 0.20 ^B^	5.80 ± 0.30 ^A^	5.99 ± 0.02 ^A^
	Lactones				
Coumarin	146.80 > 90.95	<1 µg/g ^A^	<1 µg/g ^A^	<1 µg/g^A^	<1 µg/g^A^

The results represent the means ± standard deviation. Different letters in line indicate significant differences by Tukey’s test (≤0.05).

**Table 2 pharmaceuticals-18-01229-t002:** Chemical composition of *Humulus lupulus* L. leaves crude extract (CE) from Cascade, Columbus, Comet, and Nugget varieties (µg/g).

Compound		Precursor and Productions	Cascade	Columbus	Comet	Nugget
		*m*/*z*				
		[M-H]				
	Phenolic Acids					
Clorogenic acid		354.80 > 162.95	120.079 ± 2.589 ^B^	116.728 ± 1.919 ^A^	88.510 ± 7.101 ^B^	114.786 ± 0.465 ^B^
Sinapic acid		225.00 > 206.95	13.588 ± 0.159 ^B^	68.604 ± 4.498 ^A^	15.899 ± 0.880 ^B^	60.870 ± 4.385 ^A^
Protocatechuic acid		153.20 > 108.95	150.327 ± 7.818 ^C^	440.340 ± 1.069 ^A^	198.009 ± 3.340 ^B^	154.820 ± 12.153 ^C^
*p*-Hydroxybenzoic acid		137.20 > 92.90	160.596 ± 10.599 ^C^	402.108 ± 335 ^A^	259.665 ± 13.260 ^B^	261.192 ± 13.661 ^B^
*p*-Coumaric acid		163.20 > 119.00	121.808 ± 8.819 ^B^	324.489 ± 12.106 ^A^	316.636 ± 11.398 ^A^	351.902 ± 2.624 ^A^
Caffeic acid		179.20 > 135.00	57.132 ± 2.264 ^B^	89.278 ± 2.750 ^A^	57.931 ± 3.287 ^B^	67.170 ± 5.059 ^B^
Ferulic acid		193.20 > 133.95	124.179 ± 6.323 ^B^	33.175 ± 28.197 ^A^	78.208 ± 2.748 ^B^	79.097 ± 0.992 ^B^
	Organic Acids					
Quinic acid		191.20 > 84.80	62.762 ± 2.371 ^B^	66.513 ± 0.521 ^B^	49.185 ± 2.577 ^C^	81.253 ± 3.374 ^A^
Malic acid		133.20 > 114.90	41.851 ± 0.060 ^A^	42.080 ± 1.511 ^A^	21.753 ± 1.731 ^B^	25.632 ± 0.138 ^B^
Nicotinic acid		122.40 > 77.90	113.368 ± 4.824 ^B^	175.723 ± 8.326 ^A^	53.480 ± 3.968 ^D^	76.379 ± 2.059 ^C^
	Flavonoids					
Catechin		289.10 > 245.10	ND	ND	74.206 ± 5.217 ^A^	71.339 ± 2.473 ^A^
Epicatechin		289.10 > 245.10	ND	ND	79.518 ± 6.356 ^A^	81.253 ± 3.374 ^A^
Quercetin		301.10 > 151.00	ND	ND	11.823 ± 0.972 ^A^	6.845 ± 0.024 ^B^
Rutin		609.00 > 300.00	>500 µg/g ^A^	>500 µg/g ^A^	> 500 µg/g ^A^	>500 µg/g ^A^
Luteolin		286.90 > 152.85	<1 µg/g ^A^	<1 µg/g ^A^	<1 µg/g ^A^	<1 µg/g ^A^
Naringenin		271.10 > 151.00	ND	ND	8.395 ± 0.635 ^B^	36.018 ± 1.652 ^A^
	Phenolic Aldehydes					
Hydroxybenzaldehyde		123.10 > 76.9500	33.898 ± 0.598 ^B^	117.360 ± 6.586 ^A^	119.836 ± 8.836 ^A^	109.823 ± 5.818 ^A^
Vanillin		151.40 > 136.05	102.835 ± 5.839 ^A^	196.922 ± 10.557 ^AB^	248.363 ± 22.271 ^C^	157.535 ± 1.061 ^B^
Isovaniline		153.10 > 65.00	21.776 ± 1.694 ^B^	28.420 ± 1.958 ^A^	22.678 ± 0.425 ^AB^	27.601 ± 1.398 ^AB^
Syringaldehyde		183.00 > 94.95	17.992 ± 0.985 ^B^	40.758 ± 1.093 ^A^	22.743 ± 2.206 ^B^	34.031 ± 2.649 ^A^
	Alkaloids					
Caffeine		194.90 > 137.95	16.126 ± 0.904 ^A^	18.973 ± 0.329 ^A^	102.02 ± 0.865 ^B^	18.023 ± 1.731 ^A^

The results represent the means ± standard deviation. Different letters in line indicate significant differences by Tukey’s test (≤0.05).

**Table 3 pharmaceuticals-18-01229-t003:** Total phenolic content of *Humulus lupulus* L. cones and leaves crude extract (CE) at concentrations of 1.00, 0.75, 0.50, and 0.25 (mg/mL) from Cascade, Columbus, Comet, and Nugget varieties.

	Total Phenol Content (µg Gallic Acid Equivalent/mg CE)			
	Cones			
	1.00	0.75	0.50	0.25
Cascade	95.95 ± 2.43 ^A,a,***A***^	76.33 ± 3.14 ^A,b***,A***^	54.52 ± 1.61 ^A,c,***A***^	44.23 ± 0.48 ^A,d,***A***^
Columbus	88.92 ± 7.51 ^A,a,***A***^	76.79 ± 3.34 ^A,b,***A***^	47.17 ± 3.04 ^B,c, ***B***^	23.44 ± 0.5 ^BC,d,***CD***^
Comet	76.42 ± 0.36 ^B,a, ***BC***^	52.19 ± 0.73 ^C,b***,C***^	37.90 ± 0.77 ^C, c, ***C***^	19.68 ± 1.30 ^C, d,***CDE***^
Nugget	87.48 ± 4.79 ^AB, a, ***AB***^	68.31 ± 0.39 ^B, b, ***B***^	48.54 ± 3.89 ^AB, c, ***AB***^	24.96 ± 2.70 ^B, d,***C***^
Leaves				
Cascade	53.08 ± 4.17 **^B^**^, a**, *DE***^	41.62 ± 1.93 **^B^**^, b, ***D***^	27.61 ± 1.41 ^B, c, ***A***^	13.98 ± 0.42 ^B, d, ***E***^
Columbus	46.39 ± 4.57 **^B^**^, a, ***E***^	34.90 ± 2.11 **^C^**^, b, ***E***^	29.34 ± 2.41 ^AB, b, *B*^	13.38 ± 2.03 ^B,c, ***E***^
Comet	64.36 ± 5.46 **^A^**^, a, ***CD***^	46.32 ± 2.06 **^AB^**^, b, ***CD***^	35.96 ± 4.14 ^A, c, ***CD***^	17.69 ± 2.46 ^B, d, ***DE***^
Nugget	69.70 ± 0.87 **^A^**^, a, ***C***^	49.56 ± 2.23 **^A^**^, b, ***C***^	35.16 ± 1.07 ^A, c, ***CD***^	35.66 ± 5.08 ^A, c, ***B***^

The results represent the means ± standard deviation in triplicate. Different letters indicate significant differences by Tukey’s test (≤0.05). Equal values with uppercase letters in columns (cones and leaves evaluated separately), lowercase letters in lines, and uppercase letters in italics in columns (leaves and cones evaluated together) do not differ significantly according to Tukey’s test. CE = Crude Extract.

**Table 4 pharmaceuticals-18-01229-t004:** Flavonoid content of *Humulus lupulus* L. cones and leaves crude extract (CE) at concentrations of 1.00, 0.75, 0.50, and 0.25 (mg/mL) from Cascade, Columbus, Comet, and Nugget varieties.

	Flavonoid Content (µg Equivalent of Quercetin/mg of CE)			
	1.00	0.75	0.5	0.25
Cones				
Cascade	193.82 ± 1.20 ^A.a.***E***^	188.30 ± 3.66 ^A,a,***D***^	189.33 ± 2.40 ^A,a**, *E***^	187.34 ± 6.51 ^A, a, ***F***^
Columbus	170.53 ± 2.77 ^A, a, ***F***^	163.61 ± 3.23 ^A, a, ***E***^	164.00 ± 3.25 ^A, a, ***F***^	167.87 ± 5.72 ^A, a, ***G***^
Comet	196.68 ± 2.40 ^B, b, ***E***^	193.24 ± 2.97 ^B, b, ***D***^	197.59 ± 2.86 ^B, b, ***E***^	222.21 ± 1.30 ^A, a, ***DE***^
Nugget	247.15 ± 2.11 ^AB,ab, ***D***^	247.84 ± 3.00 ^AB, ab***, C***^	255.15 ± 2.28 ^A, a, ***C***^	238.13 ± 6.51 ^B, b, ***D***^
Leaves				
Cascade	438.05 ± 2.76 **^A^**^, a, ***C***^	443.25 ± 3.42 **^A^**^, a, ***C***^	450.28 ± 7.57 **^A^**^, a, ***b***^	459.93 ± 9.86 **^A^**^,a,***c***^
Columbus	620.27 ± 10.08 **^B^**^C, b, ***B***^	610.03 ± 8.99 **^C^**^,a, ***A***^	627.42 ± 9.44 **^B^**^, a, ***a***^	649.56 ± 15.96 **^A^**^, a**, *a***^
Comet	696.67 ± 4.03 **^A^**^, a, ***A***^	602.97 ± 3.42 **^C^**^, c, ***A***^	620.86 ± 3.53 **^B^**^, b, ***a***^	627.87 ± 8.63 **^B^**^, b, ***b***^
Nugget	243.13 ± 2.28 **^A^**^, a, ***D***^	158.53 ± 2.81 **^C^**^, c, ***E***^	227.85 ± 5.53 **^B^**^, b**, *d***^	216.97 ± 7.72 **^B^**^, b, ***e***^

The results represent the means ± standard deviation in triplicate. Different letters indicate significant differences by Tukey’s test (≤0.05). Equal values with uppercase letters in columns (cones and leaves evaluated separately), lowercase letters in lines, and uppercase letters in italics in columns (leaves and cones evaluated together) do not differ significantly according to Tukey’s test. CE = Crude Extract.

**Table 5 pharmaceuticals-18-01229-t005:** Antioxidant activity determined by 2.2-diphenyl-1-picrylhydrazyl radical scavenging (DPPH^•^) and ferric reducing antioxidant power (FRAP) and 2.2′-azino-bis-(3-ethylbenzothiazoline-6-sulfonic) acid) (ABTS^•+^) assays of *Humulus lupulus* L. cones and leaves crude extract (CE) from Cascade, Columbus, Comet, and Nugget varieties.

	DPPH^•^ IC_50_ (mg/mL)	FRAP IC_50_ (µM Ferrous Sulfate/mg)	ABTS^•+^ IC_50_ (mM of Trolox/mg of CE)
	Cones		
Cascade	1.34 ± 0.16 **^B^**^,*AB*^	2.35 ± 0.03 ^C,*BC*^	5.11 ± 0.61 ^B,*B*^
Columbus	1.62 ± 0.23 **^B^**^,*AB*^	2.33 ± 0.10 ^C,*BC*^	5.40 ± 0.48 ^BC,*B*^
Comet	2.97 ± 0.59 **^C^**^,*B*^	2.52 ± 0.04 ^B,*B*^	7.10 ± 0.68 ^D,*B*^
Nugget	2.25 ± 0.83 **^BC^**^,*AB*^	2.42 ± 0.03 ^BC,*B*^	6.42 ± 0.12 ^DC,*B*^
Quercetina	0.01 ± 0.01 **^A^**^,*A*^	-	-
Trolox	-	9.17 ± 0.01 ^A^	0.25 ± 0.01 ^a,*A*^
	Leaves		
Cascade	2.87 ± 1.03 **^b^**^,*B*^	1.19 ± 0.05 ^c, *D*^	22.60 ± 1.56 ^D,*E*^
Columbus	3.13 ± 1.16 **^b^**^,*B*^	2.36 ± 0.15 ^b,*B*^	15.19 ± 0.75 ^C,*D*^
Comet	1.63 ± 1.37 **^ab^**^, *AB*^	2.15 ± 0.41 ^b, *BC*^	12.50 ± 1.06 ^B, *C*^
Nugget	2.99 ± 1.12 **^b^**^, *B*^	1.92 ± 0.07 ^b, *C*^	10.99 ± 0.75 ^B, *C*^
Quercetina	0.01 ± 0.01 **^a^**^, *A*^	-	-
Trolox	-	9.17 ± 0.01 ^a, *A*^	0.25 ± 0.01 ^a, *A*^

The results represent the means ± standard deviation in triplicate. Different letters indicate significant differences by Tukey’s test (≤0.05). Equal values with lowercase letters in columns (cones and leaves evaluated separately) and uppercase letters in italics in columns (leaves and cones evaluated together) do not differ significantly according to Tukey’s test, do not differ significantly according to Tukey’s test. IC_50_ DPPH (2.2-diphenyl-1 picrylhydrazyl); Quercetin (0.010 mg mL^−1^). CE = Crude Extract. IC_50_ = Concentration of extracts that inhibits 50% of the free radicals.

**Table 6 pharmaceuticals-18-01229-t006:** Sun protection factor (SPF) of *Humulus lupulus* L. cones and leaves crude extract (CE) (mg/mL) from Cascade, Columbus, Comet, and Nugget varieties.

	Sun Protection Factor (SPF)			
	1.00	0.75	0.50	0.25
	Cones			
Cascade	39.48 ± 0.09 ^A,a,***A***^	27.56± 2.97 ^A,b,***A***^	20.86 ± 3.29 ^A,b,***A***^	12.40 ± 3.69 ^A,c,***A***^
Columbus	33.27 ± 0.19 ^B,a,***B***^	25.20 ± 0.28 ^A,b,***A***^	15.80 ± 0.11 ^B,c,***B***^	7.53 ± 0.53 ^A,d, ***BCD***^
Comet	38.88 ± 0.25 ^A,a,***A***^	27.66 ± 0.14 ^A,b,***A***^	21.33 ± 0.43 ^A,c,***A***^	8.84 ± 0.47 ^A,d,***ABC***^
Nugget	39.07 ± 0.83 ^A,a,***A***^	28.00 ± 0.52 ^A,b,***A***^	19.41 ± 0.08 ^AB,c,**A**^	10.438 ± 0.72 ^A,d,***AB***^
	Leaves			
Cascade	16.02 ± 0.11 **^D^**^,a,***F***^	12.15 ± 0.01 **^A^**^,b,***B***^	7.81 ± 0.12 **^C^**^,c,***D***^	4.57 ± 0.29 **^A^**^,d,***DE***^
Columbus	17.90 ± 0.45 **^C^**^,a,***E***^	13.71 ± 0.26 **^A^**^,b,***B***^	8.40 ± 0.55 **^C^**^,c,***CD***^	5.38 ± 0.40 **^A^**^,d,***CDE***^
Comet	22.37 ± 0.66 **^A^**^,a,***C***^	14.90 ± 4.50 **^A^**^,b,***B***^	11.27 ± 0.16 **^A^**^,b,***C***^	3.12 ± 0.09 **^B^**^,c,***E***^
Nugget	20.45 ± 0.34 **^B^**^,a,***D***^	15.52 ± 0.36 **^A^**^,b,***B***^	10.08 ± 0.63 **^B^**^,c,***CD***^	4.72 ± 0.49 ^A**D**,d,***DE***^

The results represent the means ± standard deviation in triplicate. Different letters indicate significant differences by Tukey’s test (≤0.05). Equal values with uppercase letters in columns (cones and leaves evaluated separately), lowercase letters in lines, and uppercase letters in italics in columns (leaves and cones evaluated together) do not differ significantly according to Tukey’s test. CE = Crude Extract.

**Table 7 pharmaceuticals-18-01229-t007:** UVA/UVB ratio, anti-UVA protection, and critical wavelength (λc) of *Humulus lupulus* L. cones and leaves crude extract (CE) at concentrations of 1.00, 0.75, 0.50, and 0.25 (mg/mL) from Cascade, Columbus, Comet, and Nugget varieties.

UVA/UVB Ratio, Anti-UVA Protection, and Critical Wavelength (λc)						
	1.00	0.75	0.50	0.25	anti-UVA protection	CW(λc)
			Cones			
Cascade	1.95 ± 0.01 ^A,c,***E***^	1.93 ± 0.01 ^A,c,***D***^	1.98 ± 0.01 ^A,b,***E***^	2.02 ± 0.01 ^A,a,***A***^	Ultra	370
Columbus	1.58 ± 0.01 ^C,c,***G***^	1.58 ± 0.01 ^D,c,***F***^	1.67 ± 0.01 ^B,b,***F***^	1.72 ± 0.01 ^B,a,***B***^	Ultra	370
Comet	1.76 ± 0.02 ^B,a,F^	1.62 ± 0.01 ^C,c,***EF***^	1.64 ± 0.01 ^C,c,***F***^	1.68 ± 0.01 ^C,b,***C***^	Ultra	370
Nugget	1.73 ± 0.01 ^B,a,***F***^	1.65 ± 0.02 ^B,b,***E***^	1.67 ± 0.01 ^B,b,***F***^	1.72 ± 0.01 ^B,a,***B***^	Ultra	370
Leaves						
Cascade	2.29 ± 0.01 **^D^**^,b,*D*^	2.55 ± 0.01 **^B^**^,a,***B***^	2.22 ± 0.01 **^D^**^,c,***D***^	2.12 ± 0.02 **^C^**^,d,***A***^	Ultra	375
Columbus	2.41 ± 0.02 **^C^**^,a,***C***^	2.42 ± 0.01 **^C^**^,a,***C***^	2.42 ± 0.01 **^C^**^,a,***C***^	2.10 ± 0.01 **^C^**^,b,***B***^	Ultra	380
Comet	2.50 ± 0.01 **^B^**^,a,***B***^	2.42 ± 0.03 **^C^**^,b,***C***^	2.50 ± 0.02 **^B^**^,a,***B***^	2.41 ± 0.09 **^B^**^,c***,C***^	Ultra	380
Nugget	3.00 ± 0.01 **^A^**^,a,***A***^	3.02 ± 0.02 **^A^**^,a,***A***^	3.03 ± 0.01 **^A^**^,a,***A***^	3.02 ± 0.03 **^A^**^,b,***B***^	Ultra	385

The results represent the means ± standard deviation in triplicate. Different letters indicate significant differences by Tukey’s test (≤0.05). Equal values with uppercase letters in columns (cones and leaves evaluated separately), lowercase letters in lines, and uppercase letters in italics in columns (leaves and cones evaluated together) do not differ significantly according to Tukey’s test. CE = Crude Extract.

**Table 8 pharmaceuticals-18-01229-t008:** Degrees of anti-UVA protection according to the Boots star rating system.

UVA/UVB Limits						
	0–0.2	0.21–0.4	0.41–0.6	0.61–0.8	0.81–0.9	>0.91
Number of stars	-	*	**	***	****	*****
UVA protection	Very low	Moderate	Good	Superior	Maximum	Ultra

## Data Availability

The datasets generated during and/or analyzed during the current study are available from the corresponding author upon reasonable request.

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
