# Peer review of "Phytochemical Profile, Antioxidant Capacity, and Photoprotective Potential of Brazilian Humulus Lupulus"

_pharmaceuticals, 2025, doi:10.3390/ph18081229_

Round 1
Reviewer 1 Report
Comments and Suggestions for Authors
The manuscript needs Major revision

Author Response
We thank the reviewers for their contribution, which helped improve the manuscript's clarity and comprehension. The manuscript was modified to address the issues pointed out by the reviewers. Changes are highlighted in red in the text. We are at your disposal for any clarification.
Sincerely,
The authors
REVIEWER 1'S COMMENT
The publication titled " Phytochemical Profile, Antioxidant Capacity, and Photoprotective Potential of Brazilian Humulus lupulus" (Publication ID: pharmaceuticals-3756833) is quite intriguing. Nevertheless, in order to meet the criteria for publication in the Pharmaceuticals Journal (ISSN No: 1424-824), this work requires some enhancements. Here are some enhancements that should be taken into account:
Comments 1. The text references the aluminum chloride colorimetric method for determining flavonoid concentration (Alves and Kubota [82]). Has this procedure been suitably validated for the particular hop extracts examined, and have potential interferences from other chemicals been considered?
Response: We appreciate the reviewer's care in certifying the efficiency of the protocols used in this research. We inform the reviewer that the protocol adopted by our research group for flavonoid detection is based on the protocol established by Alves and Kubota [62]. We inform that this protocol is certified by the Brazilian Pharmacopoeia, 6th edition, and is described in the section on total flavonoid determination (Brazilian Pharmacopoeia, 2019). For more information, follow the link to access the Brazilian Pharmacopoeia. (https://bibliotecadigital.anvisa.gov.br/jspui/bitstream/anvisa/540/1/Cal%C3%AAndula%2C%20flor_PM022-01_6ed_2019.pdf).
Comments 2. The research utilizes ABTS, FRAP, and DPPH tests. Does it address the complimentary characteristics of these tests and provide a rationale for their selection in assessing the antioxidant activity of hop extracts?
Response to reviewer: The selection of these methods is based on the identification of phenolic compounds in the samples (Tables 1 and 2), and according to the literature, the primary mechanism of action of phenolic antioxidants involves the neutralization of stable radical species. This neutralization is assessed by the ABTS [20] and DPPH [28] methods, and by the reduction of ferric ions, evaluated by the FRAP [29] method. This explanation can be found in the discussion section, lines 312 to 341.
Comments 3. The manuscript indicates that protocatechuic acid was present in significant quantities in Nugget cone extracts. Are the quantification methodologies and detection thresholds explicitly delineated, and how do these quantities relate to other hop varieties or values in the literature?
Response to reviewer: The figure below shows the print of the report generated by Insight Software (Shimadzu®), regarding protocatechuic acid for all samples analysed in this study. This acid was confirmed by all precursor ions (blue and green peaks) detected by the UHPLC system.
PS: ‘folha’ means leaves.
Responses to Reviewer 1, including the figures and graphs, have been added to the end of the manuscript.
Comments 4. The presence of morin is exclusively noted in Nugget (263.37 µg/g) and Cascade extracts. Is there an examination of the absence of morin in other kinds, and are potential biosynthetic pathways or environmental influences investigated?
Response: We appreciate the reviewer's comment. According to Table 1, the flavonoid Morin was identified in the cones of the four varieties Cascade (150.76 ± 9.48B), Columbus (86.94 ± 2.78C), Comet (141.04 ± 1.58B), Nugget (265.37 ± 10.28A). Higher levels were found in the Nugget variety, and equal levels in the Cascade and Comet varieties. Lower concentrations were found in the Columbus variety. The probable explanation for this difference in Morin concentrations may be in the studies conducted by Mondal et al. (2022). These authors state that the genetic architecture of the morin biosynthetic pathway is still unclear. However, to understand the genetic architecture of the morin biosynthetic pathway, the following components were analyzed: (1) cis-responsive element (CRE)-mediated regulation, (2) microRNAs (miRNA)-mediated post-transcriptional silencing, and (3) tissue-specific in silico gene expression. The data highlighted the morin biosynthetic pathway-associated genes, namely, phenylalanine ammonia-lyase (MnPAL), chalcone synthase A (MnCSA), chalcone-flavonone isomerase (MnCFI), and flavonoid 3′,5′-hydroxylase (MnFH), which are transcriptionally regulated by different growth, development, and stress-responsive CREs.
Mondal, R., Antony, S., Thriveni, M. C., Thanavendan, G., Ravikumar, G., & Sreenivasa, B. T. (2022). Genetic architecture of morin (pentahydroxyflavone) biosynthetic pathway in mulberry (Morus notabilis): an in silico approach. Journal of Berry Research, 12(4), 483-494.https://doi.org/10.3233/JBR-220032.
We would like to point out that this likely explanation for the morin levels was not included in the text, because when comparing the morin levels in the studied varieties, we would also have to evaluate the other identified compounds.
Comments 5. The calibration curve for gallic acid equivalents (Equation 1) is referenced. Is the coefficient of determination (R²) provided, and is it sufficiently elevated to provide reliable quantification of phenolic content?
Response: We appreciate the reviewer's comment. This is an important point to highlight: the calibration curve is presented in Equation 1 in the Materials and Methods section, and it had an R2 value of 0.9998. Therefore, supporting the reliability of the test results.
Comments 6. The study references Biskup et al. [20] regarding the efficacy of phenolic compounds containing hydroxyl groups in ortho or para configurations for ABTS radical scavenging. Does the study furnish experimental data to validate this association for the specified components in hop extracts?
Response: We appreciate the reviewer's comment. Yes, the Biskup et al. [20] study provides experimental data. Corroborating these findings, Biskup et al. [20] investigated the antioxidant potential (IC50) of 10 water-soluble phenolic compounds using the ABTS and FRAP methods. The authors observed that phenols with a greater number of hydroxyl groups had greater antioxidant capacity, as indicated by lower IC50 values, 4.33 mM and 5.23 mM, respectively, for gallic acid and pyrogallol. Another finding in this experiment was that the type of spacer between the carboxylic acid and the benzene ring influences its activity, which increases for the methylene group in phenolic acids. They also found that phenols with two hydroxyl groups attached to the aromatic ring in the ortho position suppress the ABTS•+ radical more strongly than compounds with hydroxyl groups substituted in the meta position.
Comments 7. The term "Sun Protection" is used, although the document does not provide an indepth analysis of this application. Does the publication present experimental evidence, such as SPF measurements, to substantiate the claims regarding the sun protection capability of Brazilian hop extracts?
Response: We thank the reviewer for their observations. The manuscript presents experimental evidence supporting the photoprotective potential of H. lupulus extracts. In the Methods and Results sections, we showed the in vitro SPF using the method proposed by Mansur, with results ranging from 16.02 to 39.48 depending on the plant organ and hop variety (Table 6). Additionally, we evaluated the UVA/UVB ratio, anti-UVA protection factor, and critical wavelength (λc), as presented in Table 7. The values obtained, as λc ≥ 370 nm and UVA/UVB ratio > 0.9, agree with the international criteria for “broad-spectrum” UV protection, and they are classified as “ultra protection” based on the Boots star rating system.
The analytical parameters used in this study, such as SPF, UVA/UVB ratio, anti-UVA protection factor, and critical wavelength (λc), are widely employed in the scientific literature to assess the photoprotective potential of natural products and formulations (August et al., 2024; Castro et al., 2023; Quintero-Rincón et al., 2023) and the results are accepted as a measure of photoprotection.
In the discussion section, we presented a critical argumentation about the photoprotective potential results, and we correlated them with the chemical compositions of the extracts. Particular emphasis was placed on the presence and relative abundance of phenolic compounds and flavonoids, which are known to contribute to UV absorption and antioxidant defense. These aspects were addressed from line 439 to line 452.
August, E. M. G., Horn, P. A., Cavichioli, N., Rebelo, A. M., Reinke, C. K., & Zeni, A. L. B. (2025). Seasonal Phenolic Profile, Antioxidant, and Photoprotective Activities of Psidium guajava L. Leaves. Chemistry & Biodiversity, 22(5), e202402852.
Castro, T. L. A., Souza, L. P., Lima-Junior, S. E., & Cardoso, C. A. L. (2023). Optimization of obtaining extracts with photoprotective and antioxidant potential from Campomanesia adamantium (Cambess.) O. Berg. Sustainable Chemistry and Pharmacy, 31, 100945.
Quintero-Rincón, P., Mesa-Arango, A. C., Flórez-Acosta, O. A., Zapata-Zapata, C., Stashenko, E. E., & Pino-Benítez, N. (2023). Exploring the potential of extracts from Sloanea medusula and S. calva: formulating two skincare gels with antioxidant, sun protective factor, and anti-Candida albicans activities. Pharmaceuticals, 16(7), 990.
Comments 8. Does the manuscript juxtapose the antioxidant or phenolic content of Brazilian hops with that of non-Brazilian hop cultivars or alternative plant sources to explain its findings?
Response: We appreciate the reviewer's comment. Lines 160 to 172 contain information regarding the difference in total phenol and flavonoid content of hops and the same varieties grown in Brazil and other countries.
“Previous studies in the literature further support these findings. Lela et al. [17], in a study conducted at EVRA srl (Galdo di Lauria, Italy), also reported the presence of various polyphenolic compounds in hop extracts. Vergun et al. [18] analyzed different parts of the hop plant and reported elevated polyphenol levels during the flowering stage in ethanolic extracts, with values ranging from 23.76 to 54.13 mg GAE/g. Together, these results corroborate the high total phenol content observed in the present study (Table 3). The flavonoid and total phenol content was investigated by Almeida et al. [8], with hop cones of the Cascade variety grown in Brazil (Itu-SP) and in Washington (USA). The two materials were subjected to antioxidant tests, indicating Flavonoids (BR: 54.470 μg/mL; USA: 52.940 μg/mL) and in our research (193.82 μg/mL). The Total Phenols content was (BR: 33.930 μg/mL; USA: 27.310 μg/mL QE), and in our research (95.95 μg/mL QE). The difference found can be justified by edaphoclimatic factors, since it is the same variety.”
Comments 9. Considering the emphasis on phenolic aldehydes and flavonoids, does the manuscript address potential medicinal or cosmetic applications of the hop extracts, substantiated by the data provided?
Response: We appreciate the reviewer's comment. The following information was inserted in the discussion of results (lines 498-512):
Hop plants are almost exclusively cultivated for the brewing industry, as the resins and essential oils from the female cones are used to impart flavor to beer [61]. However, studies on the four hop varieties have shown that they also contain flavonoids and phenolic aldehydes in both the cones and leaves (Tables 1 and 2), indicating potential industrial applications beyond beer production.
In cosmetics, flavonoids are valued for their sunscreen, anti-aging, and anti-inflammatory properties. In the food industry, these compounds help extend the shelf life of foods thanks to their antimicrobial and antioxidant properties [62]. As for phenolic aldehydes, such as dehydroxybenzaldehydes in ortho, para, and meta positions (o-, p-, and m-hydroxybenzaldehydes), they are of great importance both in biology and in industry, as they participate in proton transfer discussions, essential in processes ranging from acid-base reactions to enzyme-catalyzed solutions [63]. Finally, it is worth highlighting vanillin, a phenolic aldehyde widely used as a flavoring in food and cosmetics, due to its sensory and functional properties [64].
Comments 10. The publication details the concentration of hop extracts under reduced pressure utilizing a rotary evaporator (Tecnal TE-211). Does it assess the efficacy of the extraction process (e.g., yield percentages) and examine how extraction parameters (e.g., solvent type, temperature) may influence the recovery of certain phenolic compounds such as rutin or protocatechuic acid?
Response: We appreciate the reviewer's comment. Information on the extract yield (%) found in Figure 1 (Lines 122-126) was entered. The choice of the extraction method (maceration) and the solvent used (96ºG.L ethyl alcohol) was based on previous studies with hops (Lela et al., [18]. The research did not aim to evaluate the influence of other extraction methods, solvents, and temperature on the recovery of rutin and protocatechuic acid. These analyses are already planned for future studies.
Comments 11. The study indicates variations in antioxidant activity among hop types, such as Saaz, Cascade, and Vloria. Are these differences statistically significant, and does the manuscript investigate potential genetic or biochemical mechanisms influencing varietal-specific phenolic profiles?
Response: We appreciate the reviewer's comment. The varieties investigated in this study were Cascade, Columbus, Comet, and Nugget. We can see that there was a significant difference in the antioxidant response using the DPPH, ABTS, and FRAP methods, as shown in Table 5, which is supplemented in lines 257-280.
“No less important than the antioxidant protocols used is the influence of genetic variability associated with climatic factors such as sun exposure, temperature, and water availability on the chemical composition of H. lupulus cultures [10]. In the present study, the impact of these variables on the antioxidant response was observed, as shown in Table 5, indicating a significant difference in the antioxidant response of the four varieties investigated. For optimal growth, hop plants need particular climatic conditions such as exposure to low temperatures during dormancy, mild temperatures in spring, sufficient moisture from irrigation or rainfall throughout the season, and dry weather during harvest time [3]. In this sense, Foster et al. [27] evaluated the impact of climatic conditions on the biogenesis of various compounds in hops in European hop-producing countries. The authors assert that high temperatures and low moisture content are the factors that most negatively impact hop production and its metabolites. They report in their studies that the compounds most sensitive to these parameters are α-acids, with a 30% decrease in production, followed by a 17% decrease in essential oil synthesis and phenolic compounds, with a 9% sensitivity level.
Therefore, we consider that the total phenol and flavonoid contents found in the present experiment (Tables 3 and 4) may also have been affected by high temperatures (27.09 ± 4.00 °C) from November 2022 to January 2023, as well as low humidity reflected by the low rainfall (5,184 ± 10.04 mm). Another factor that may have affected the chemical composition and interfered with the antioxidant response is the soil where the hops are grown: sandy texture, low water retention capacity, and low organic matter content [57]. According to Sposito et al. [58], hop plants grow well in different types of soil, as long as they are fertile and able to retain moisture.”
Comments 12. The ABTS assay values for leaf extracts vary from 10.987 to 22,600 µM Trolox equivalents per milligram of extract. Does the text examine the extensive diversity in these numbers and consider probable causes of error or biological variability in the samples?
Response: We appreciate the reviewer's comment. We inform the reviewer that the antioxidant results by the ABTS method were changed from µM to mM. The positive control Trolox (0.25 mM of Trolox/mg of CE) (Table 5) was also included for better comparison and interpretation of the results. Our results were compared to two studies: one conducted in South Korea and the other in Brazil, which showed inferior results, justified by the influence of climatic factors and location, since in Brazil, the experiment was conducted in two regions considered ideal for hop cultivation, due to their climate and altitude. Lines 319-341.
Comments 13. Considering the emphasis on polyphenols, does the study evaluate the stability of the found chemicals (e.g., protocatechuic acid, morin) during storage or processing, and what implications would this have for their prospective therapeutic applications?
Response: We thank the reviewer for the comment. The present study did not evaluate the chemical stability of polyphenols during storage or processing. However, we agree that stability is a critical aspect when considering the therapeutic or cosmetic application of bioactive compounds. We have added a sentence in the discussion to recognize this limitation and emphasize the need for future studies on the stability of these extracts under various environmental and formulation conditions. A paragraph on the importance of evaluating the stability of phenolic compounds was included in the results and discussion session on lines 461 to 467.
Results and Discussion lines 513-519: Although the extracts demonstrated significant levels of bioactive polyphenols, the present study did not assess their chemical stability during storage or processing. Phenolic compounds are known to be sensitive to pH, light, temperature, and oxygen exposure [70], which can influence their bioactivity and shelf life. Therefore, future studies should investigate the stability of these compounds in various delivery systems and under different storage conditions to ensure efficacy and formulation compatibility in prospective therapeutic applications.
Comments 14. Does the publication provide dose-response curves for the antioxidant assays (DPPH, FRAP, ABTS) to illustrate the correlation between extract concentration and antioxidant activity, or is the data restricted to single-point measurements?
Response: We appreciate the reviewer’s comment. Yes, for the antioxidant assays (DPPH, ABTS, FRAP), multiple concentrations (1.0, 0.75, 0.50, and 0.15 mg/mL) were tested, and the dose–response curves were constructed. The IC₅₀ values were calculated from the linear regression equation fitted to these data. The corresponding curves and equations will be presented as supplementary material.
Responses to Reviewer 1, including the figures and graphs, have been added to the end of the manuscript.
Comments 15. The total phenolic content was quantified using a SpectraMax Plus 384 Microplate Reader at a wavelength of 760 nm. Does the study address potential interferences from non-phenolic chemicals in the hop extracts that may influence absorbance readings and result in an overestimation of phenolic content?
Response: We appreciate the reviewer’s comment. This methodology, initially proposed by Swain and Hillis [60], modified by Sá [61], does not evaluate only phenolic compounds; instead, it measures the total reducing capacity of the sample. Although phenolic content is the main contributor, other classes of molecules can also influence the results. For this reason, the method expresses the result in acid gallic equivalent. Nevertheless, this methodology is widely accepted for the quantification of total phenolics.
Comments 16. The document indicates that hop cultivation commenced in November 2022 with 204 seedlings. Does it examine how environmental factors (e.g., soil composition, climate, irrigation) in Brazil may affect the phenolic and flavonoid content in comparison to conventional hop-growing locations such as the United States or Germany?
Response: We appreciate the reviewer's suggestion. A paragraph has been added (lines 257-272) and (lines 273-280), highlighting the influence of climatic factors on the antioxidant response.
No less important than the antioxidant protocols used is the influence of genetic variability associated with climatic factors such as sun exposure, temperature, and water availability on the chemical composition of H. lupulus cultures [10]. In the present study, the impact of these variables on the antioxidant response was observed, as shown in Table 5, indicating a significant difference in the antioxidant response of the four varieties investigated. For optimal growth, hop plants need particular climatic conditions such as exposure to low temperatures during dormancy, mild temperatures in spring, sufficient moisture from irrigation or rainfall throughout the season, and dry weather during harvest time [3]. In this sense, Foster et al. (2021) [27] evaluated the impact of climatic conditions on the biogenesis of various compounds in hops in European hop-producing countries. The authors assert that high temperatures and low moisture content are the factors that most negatively impact hop production and its metabolites. They report in their studies that the compounds most sensitive to these parameters are α-acids, with a 30% decrease in production, followed by a 17% decrease in essential oil synthesis and phenolic compounds, with a 9% sensitivity level.
Therefore, we consider that the total phenol and flavonoid contents found in the present experiment (Tables 3 and 4) may also have been affected by high temperatures (27.09 ± 4.00 °C) from November 2022 to January 2023, as well as low humidity reflected by the low rainfall (5,184 ± 10.04 mm). Another factor that may have affected the chemical composition and interfered with the antioxidant response is the soil where the hops are grown: sandy texture, low water retention capacity, and low organic matter content [57]. According to Sposito et al. [58], hop plants grow well in different types of soil, as long as they are fertile and able to retain moisture.
Comments 17. The research underscores the promise of hop-derived polyphenols for topical use. Does it address the bioavailability or dermal absorption of these substances (e.g., flavonoids, phenolic acids) to substantiate their effectiveness in cosmetic or medicinal formulations?
Response: Dear reviewer, thank you for your question. A justification for the use of polyphenols in dermal formulations was added to lines 513-519.
"Although the extracts demonstrated significant levels of bioactive polyphenols, this study did not evaluate their chemical stability during storage or processing. Phenolic compounds are known to be sensitive to pH, light, temperature, and oxygen exposure [70], which can influence their bioactivity and shelf life. Therefore, future studies should investigate the stability of these compounds in various delivery systems and under different storage conditions to ensure the efficacy and compatibility of the formulation in potential therapeutic applications."
However, the literature indicates that Humulus lupulus extracts have already been tested topically in humans, to investigate their anti-inflammatory potential in the UVB erythema test, use lipid-rich vehicles whose main components were humulone and lupulone. In the literature, there are promising results in anti-inflammatory testing using an oil-and-water cream with hop extract, but these still require further study (Hurth et al., 2022). Studies conducted by Cleemput et al. (2009) indicated that the combination of hop bitter acids resulted in additive inhibition of NF-κB activity after treatment with TNF-α. Topical application has also been tested in mice, inhibiting acute inflammation in vivo.
Hurth Z, Faber ML, Gendrisch F, Holzer M, Haarhaus B, Cawelius A, Schwabe K, Schempp CM, Wölfle U. The Anti-Inflammatory Effect of Humulus lupulus Extract In Vivo Depends on the Galenic System of the Topical Formulation. Pharmaceuticals (Basel). 2022 Mar 14;15(3):350. doi: 10.3390/ph15030350. PMID: 35337147; PMCID: PMC8951350.
Cleemput MV, Heyerick A., Libert C., Swerts K., Philippé J., Keukeleire DD, Haegeman G., Bosscher KD Bitter acids from hops efficiently block inflammation independently of GRα, PPARα, or PPARγ. Mol. Nutr. Food Res. 2009;53:1143–1155. doi: 10.1002/mnfr.200800493
Comments 18. Are suitable controls (e.g., synthetic antioxidants such as Trolox or alternative plant extracts) incorporated in the antioxidant tests to evaluate the efficacy of hop extracts, and how do the hop extracts differ regarding potency?
Response: Thank you for your comment. In the antioxidant assays, appropriate controls were included: Trolox was used in the ABTS and FRAP assays, and quercetin in the DPPH assay, all of which are widely accepted as reference antioxidants. The antioxidant activity of the H. lupulus extracts was expressed in mM Trolox equivalents (ABTS) and as EC₅₀ values (DPPH), allowing comparative evaluation of their potency across varieties and plant organs. Although no comparisons with alternative plant extracts were done during the assays, the values of antioxidant activity of H. lupulus extracts are consistent with or superior to those reported for several other polyphenol-rich botanicals in the literature.
Comments 19. The manuscript discusses the structural influences of phenolic compounds on antioxidant activity, such as electron-donating groups. Does it offer mechanistic insights, including computational modeling or experimental evidence, to elucidate how individual chemicals in hop extracts interact with free radicals in the ABTS or DPPH assays?
Response: We thank the reviewer for this observation. The manuscript discusses structure–activity relationships of phenolic compounds in general terms, based on known chemical characteristics and literature data. For instance, we highlighted how hydroxyl group position and aromatic conjugation influence antioxidant potential, particularly for compounds such as quercetin, morin, and protocatechuic acid. However, the study did not include computational modeling or direct mechanistic experiments to elucidate the molecular interactions. We agree that such approaches could provide deeper mechanistic insights, and we intend to do this analysis in future research.

Reviewer 2 Report
Comments and Suggestions for Authors
- Authors performed antioxidant potential test of the extract. However, they did not assess any parameter of lipid peroxidation, hydrogen peroxide or nitric oxide. If possible, include some of these additional experiments.
- All results are presented as Tables, looking poor. Please, improve the presentation of results by adding some graphs or schemes, images.
- Please, describe statistical analysis in more details, includin the normality of data distribution tests used.
- What was the extrat yield? add, please. Also, add the voucher number of the plant
- Please, include the lmitations of the study at the end of discussion section and enrich the discussion section by commenting of potential molecular mechsnisms of antioxidant and photoprotective properties of the investigated plant extract.
English grammar should be improved by a native speaker.
Author Response
We thank the reviewers for their contribution, which helped improve the manuscript's clarity and comprehension. The manuscript was modified to address the issues pointed out by the reviewers. Changes are highlighted in red in the text. We are at your disposal for any clarification.
Sincerely,
The authors
REVIEWER COMMENTS 2
Comments 1. Authors performed antioxidant potential test of the extract. However, they did not assess any parameter of lipid peroxidation, hydrogen peroxide or nitric oxide. If possible, include some of these additional experiments.
Response: We appreciate the reviewer’s suggestion. The present study focused on evaluating the antioxidant capacity of H. lupulus crude extracts through established chemical assays that capture electron-transfer and radical scavenging potential. We did not assess additional biological parameters such as lipid peroxidation, hydrogen peroxide scavenging, or nitric oxide inhibition in this work. However, we agree that such assays would enrich the functional characterization of the extracts. These analyses are already planned for future studies involving essential oils obtained from Brazilian H. lupulus, which may present distinct antioxidant profiles.
Comments 2. nts All results are presented as Tables, looking poor. Please, improve the presentation of results by adding some graphs or schemes, images.
Response: We appreciate the reviewer's suggestion. However, because there are so many variables being evaluated, we believe that a table format best represents the information contained.
Comments 3. Please, describe statistical analysis in more details, including the normality of data distribution tests used.
Response: Suggestion accepted. A complement was inserted in lines 707-708.
“Data normality was assessed using the Shapiro-Wilk test and the Bartlett test to assess homogeneity.”
“Data normality was assessed using the Shapiro-Wilk test and the Bartlett test to assess homogeneity.”
Comments 4. nts What was the extrat yield? add, please. Also, add the voucher number of the plant
Response: We thank the reviewer for their observations. Data regarding the botanical identification of the species H. lupulus were included in sessions 4.1 and 4.2 of the methodology and Results and discussion, and the yield data (%) are presented in the form of a graph, represented by Figure 1.
4.1. Plant Material and Field Conditions
Lines 538-542: Botanical identification was carried out, and specimens were deposited in the Herbarium of the Universidade Paranaense under number 340. The species was documented in the National System of Genetic Heritage Management and Associated Traditional Knowledge (SisGen, as abbreviated in Portuguese) with identification number AC98839.
4.2. Extract obtention
Lines 559-561. The yield (%) of the extracts was calculated from the ratio of the dry mass of the plant material (g) divided by the mass of the crude extract (g).
Results and Discussion
Lines 122-123: The yield of crude extracts from cones and leaves of the four varieties of H. lupulus is shown in Figure 1, indicating higher yields for cones when compared to the leaves.
Figure 1: Yield (%) of crude extract from cones and leaves of Humulus lupulus varieties Cascade, Columbus, Comet, and Nugget.
Comments 5. Please, include the lmitations of the study at the end of discussion section and enrich the discussion section by commenting of potential molecular mechsnisms of antioxidant and photoprotective properties of the investigated plant extract.
Response: We appreciate the reviewer's suggestion. A paragraph about the limitations of this study and future projections has been added to the discussion section. Lines 470 to 476.
“Although the extracts demonstrated significant levels of bioactive polyphenols, the present study did not assess their chemical stability during storage or processing. Phenolic compounds are known to be sensitive to pH, light, temperature, and oxygen exposure [70], which can influence their bioactivity and shelf life. Therefore, future studies should investigate the stability of these compounds in various delivery systems and under different storage conditions to ensure efficacy and formulation compatibility in prospective therapeutic applications.”

Reviewer 3 Report
Comments and Suggestions for Authors
In the manuscript entitled “Phytochemical Profile, Antioxidant Capacity, and
Photoprotective Potential of Brazilian Humulus lupulus” by Canonico Silva et al.,
the phytochemical composition, antioxidant activity and photoprotective ability of four H. lupulus varieties, Cascade, Columbus, Comet, and Nugget; cultivated in the
northwestern region of Paraná State, Brazil.
The manuscript contains many valuable results, however, it requires some revision The following comments should be taken into account in the revised version of manuscript:
- The introduction should contain concise information about the scientific background of the problem, some information about the material being studied and, above all, a clearly stated goal of the work and its novelty. What distinguishes, in what way is the presented work different from other works on a similar topic.
- Why don't the authors provide the exact content of rutin in the extracts, or is it impossible to determine it precisely?
- In Table 1, capital letters “G” appear instead of “g”
- Why were polyphenol and flavonoid contents determined for different extract concentrations (Tables 3 and 4)? What was the point of this, since it is known that the higher the extract concentration, the higher the concentration of the compounds determined.
- Table 4- The authors should explain (or at least try to explain) why the flavonoid content does not change unidirectionally with the change in the extract concentration (the concentration decreases, and the content alternately decreases and increases - what could be the reason for this?
- Table 5 - why do the authors express antioxidant properties in different units? Wouldn't it be better to convert the results obtained in each method to a standard antioxidant, e.g., Trolox? It is known that it would not change the correlation between the extracts but would facilitate the interpretation of the results.
- Lines 279-281 -The sentence should be rephrased. It's difficult to conclude that the lower the IC50 value, the better the antioxidant properties.
- The discussion of the results in many places is too extensive (overly wordy), should be modified because contains too obvious information (e.g. the authors write extensively about the structure of compounds)
- For better reading and understanding of the text, the discussion should be divided into subsections (e.g. regarding the analysis of phytochemicals, separately for the assessment of antioxidant properties and photoprotective properties.
- In the experiment - there is no subsection regarding the reagents used
- Section 4.3. Since calibration curves were prepared for all the compounds mentioned, would it not be worthwhile to present their equations and main characteristics in the supplementary material?
- Section 4.6 and data in table 5 - lack of consistency in markings - IC50 or EC50
- Sections 4.6.2-4.6.3 The Authors mention that the antioxidant properties are determined for different concentrations of extracts, and for what exact concentration the data presented in Table 5 refer.
- Conclusions should be revised. This section of the manuscript should include specific information regarding the results obtained.
Author Response
We thank the reviewers for their contribution, which helped improve the manuscript's clarity and comprehension. The manuscript was modified to address the issues pointed out by the reviewers. Changes are highlighted in red in the text. We are at your disposal for any clarification.
Sincerely,
The authors
REVIEWER COMMENTS 3
In the manuscript entitled “Phytochemical Profile, Antioxidant Capacity, and
Photoprotective Potential of Brazilian Humulus lupulus” by Canonico Silva et al.,
the phytochemical composition, antioxidant activity and photoprotective ability of four H. lupulus varieties, Cascade, Columbus, Comet, and Nugget; cultivated in the
northwestern region of Paraná State, Brazil.
The manuscript contains many valuable results, however, it requires some revision The following comments should be taken into account in the revised version of manuscript:
Comments 1. The introduction should contain concise information about the scientific background of the problem, some information about the material being studied and, above all, a clearly stated goal of the work and its novelty. What distinguishes, in what way is the presented work different from other works on a similar topic.
Response: We appreciate the reviewer's suggestion. A paragraph was added highlighting the climatic and technological challenges faced in Brazil to ensure that hop cultivation can be consolidated, economically viable, and, most importantly, maintain the same chemical characteristics as imported hops (lines 75 to 81). The introduction also addressed climatic challenges, particularly the photoperiod required for hop production (lines 63 to 68).
The reviewer questions what distinguishes our research from other established hop research. The differences between this study and others are outlined in the objectives of the study (lines 108 to 112) and in the discussion, which addresses the local challenges of soil, temperature, and low humidity characteristic of the region where the research is being conducted (lines 257 to 282).
Comments 2: Why don't the authors provide the exact content of rutin in the extracts, or is it impossible to determine it precisely?
Response to the reviewer: In all dilutions of the crude extracts tested in the UHPLC analysis, the peak area was over the maximum concentration of the calibration curve for rutin, so it was not possible to determine its concentration precisely. Thus, it was expressed as >500 ug/g, which is usually observed in some works.
Comments 3. In Table 1, capital letters “G” appear instead of “g”
Response: We appreciate the reviewer's observation. The necessary corrections were made.
Comments 4. Why were polyphenol and flavonoid contents determined for different extract concentrations (Tables 3 and 4)? What was the point of this, since it is known that the higher the extract concentration, the higher the concentration of the compounds determined.
Response: We appreciate the reviewer's question. For the FRAP, ABTS, and DPPH methodologies, it is mandatory to use more than one concentration to generate the linear regression necessary to define the IC₅₀ of the samples. Similarly, in sunscreen assays, it is important to work with different concentrations to highlight the variation in the Sun Protection Factor (SPF) as a function of concentration, in addition to enabling the calculation of the UVA/UVB ratio and the critical wavelength. Based on this principle, for the sake of methodological consistency, we chose to use the same concentrations in all assays. This ensures that comparisons between antioxidant and photoprotective activity are consistent and reliable.
Comments 5. Table 4- The authors should explain (or at least try to explain) why the flavonoid content does not change unidirectionally with the change in the extract concentration (the concentration decreases, and the content alternately decreases and increases - what could be the reason for this?
Response: In the protocol established by our research group, we standardized concentrations of 1.0, 0.75, 0.5, and 0.25 mg/mL for all antioxidant assays, including the determination of total phenols and flavonoids.
For the determination of total phenols, a unidirectional change was observed with changes in concentration. However, when determining the total flavonoid content, this change was not observed. It was found that higher concentrations yield lower levels. We have not yet been able to find an answer for this non-unidirectional variation. However, we believe the likely explanation is the sensitivity of the method, despite its robustness and the fact that the Brazilian Pharmacopoeia recommends it.
Comments 6. Table 5 - why do the authors express antioxidant properties in different units? Wouldn't it be better to convert the results obtained in each method to a standard antioxidant, e.g., Trolox? It is known that it would not change the correlation between the extracts but would facilitate the interpretation of the results.
Response: We appreciate the editor's comment. In future studies, we may consider converting the results obtained by each method to a single antioxidant standard, as suggested. However, the current method for expressing antioxidant results in studies conducted by our research group, is based on the methodologies presented in this paper. In the methodology section, we describe in detail how the results were expressed and the protocol used, ensuring the consistency and reliability of our data.
DPPH: Based on the correlation between absorbance and concentration of the antioxidant sample, the concentration needed to reduce 50% of the free radicals (IC50) was determined. Rufino et al. [63]
FRAP: The antioxidant activity was expressed as μM ferrous sulphate/mg of sample. Rufino et al. [65]
ABTS: The results were expressed as mM of Trolox per gram of extract. Rufino et al. [66]
Flavonoids: The results were expressed in µg quercetin equivalents (EQ) per mg of extract.Alves and Kubota [62].
Phenol content (TP): The results were expressed as µg of gallic acid equivalent (GAE) per mg of sample. Swain and Hillis [60] with modifications by Sá [61].
Comments 7. Lines 279-281 -The sentence should be rephrased. It's difficult to conclude that the lower the IC50 value, the better the antioxidant properties.
Response: We appreciate the reviewer's comment. The discussion has been reworded, and the changes are outlined in red.
Comments 8. The discussion of the results in many places is too extensive (overly wordy), should be modified because contains too obvious information (e.g. the authors write extensively about the structure of compounds)
Response: We appreciate the reviewer's comment. We structured the Discussion section to (i) compare our results with similar studies in the literature—thereby situating and justifying our findings, and (ii) detail the structures of the compounds as a mechanistic basis for the observed antioxidant activity.
Comments 9. For better reading and understanding of the text, the discussion should be divided into subsections (e.g. regarding the analysis of phytochemicals, separately for the assessment of antioxidant properties and photoprotective properties.
Response: We thank the reviewer for their observations. However, the discussion is structured taking into account the influence of chemical composition on the observed antioxidant response. Our goal was to keep the text sequentially aligned with the tables and figures, facilitating comprehension. We believe that dividing the discussion into subsections could compromise the flow of information, which could make understanding more confusing.
Comments 10. In the experiment - there is no subsection regarding the reagents used
Response: We thank the reviewer for their observations. The journal’s guidelines do not require this information; therefore, a specific section was not included. Despite this, the reagents are described in the specific methodologies in which each one was used.
Comments 11. Section 4.3. Since calibration curves were prepared for all the compounds mentioned, would it not be worthwhile to present their equations and main characteristics in the supplementary material?
Response: The calibration curves were prepared by using a mix of the standard compounds, in the concentrations of 10–250 µg/L, as described in line 579 of the paper. Table 1S was included with the equation of analytical curves and their determination coefficient. An explanation about this addition was added to the text (lines 586-587).
Comments 12. Section 4.6 and data in table 5 - lack of consistency in markings - IC50 or EC50
Response: We thank the reviewer for the point. There was indeed a mistake made while writing the paper, and it has already been corrected.
Comments 13. Sections 4.6.2-4.6.3 The Authors mention that the antioxidant properties are determined for different concentrations of extracts, and for what exact concentration the data presented in Table 5 refer.
Response: We appreciate the reviewer’s comment. For Table 5, data obtained from the DPPH, ABTS, and FRAP assays were used. These assays are performed at all tested concentrations, generating a linear regression equation that yields a single value. The DPPH and ABTS results correspond to the IC₅₀ values, which represent the extract concentration (µg/mL) required to inhibit 50% of the free radicals present in the assay. Lower IC₅₀ values indicate higher antioxidant activity. The FRAP results assess the sample’s ability to reduce ferric ions (Fe³⁺) to ferrous ions (Fe²⁺); higher FRAP values indicate greater reducing power.
Comments 14. Conclusions should be revised. This section of the manuscript should include specific information regarding the results obtained.
Response: We appreciate the reviewer’s comment. The conclusions were rewrited.
These findings indicate that even with Brazil's challenging climate aspects, the Cascade, Columbus, Comet, and Nugget varieties of Humulus lupulus grown in the northwestern region of Paraná State, Brazil, possess a rich and well-balanced chemical composition, including compounds such as phenolic and organic acids, flavonoids, phenolic aldehydes, alkaloids, and α-benzopyrone lactones. Of note is the presence of rutin (>500 µg/g) in the cones and leaves of all four varieties. Furthermore, the extracts demonstrated antioxidant and photoprotective potential, which opens the door to new applications in sectors such as cosmetics, food, and even sun protection. These results encourage continued research to explore the potential of these varieties further, promoting the sustainable and innovative use of hops in Brazil.

Round 2
Reviewer 1 Report
Comments and Suggestions for Authors
The revised version of manuscript was accepted for publication
Reviewer 2 Report
Comments and Suggestions for Authors
The authors have successfully addressed most of previous comments, demonstrating thorough revisions and attention to detail. I recommend the manuscript for acceptance.
Reviewer 3 Report
Comments and Suggestions for Authors
The manuscript has been revised, and most of my comments have been addressed. However, I still believe that dividing the discussion into subsections would have made the manuscript easier to read.